# Prediction of causal genes at GWAS loci with pleiotropic gene regulatory effects using sets of correlated instrumental variables

**Mariyam Khan**[1], **Adriaan-Alexander Ludl**[1], **Sean Bankier**[1], **Johan L. M. Björkegren**[2,3], **Tom Michoel**[1]*

**1** Computational Biology Unit, Department of Informatics, University of Bergen, Bergen, Norway, **2** Department of Medicine (Huddinge), Karolinska Institutet, Huddinge, Sweden, **3** Department of Genetics & Genomic Sciences/Institute of Genomics and Multiscale Biology, Icahn School of Medicine at Mount Sinai, New York, New York, United States of America

* Tom.Michoel@uib.no

**Data Availability Statement:** All source code related to the paper is available at https://github.com/mariyam-khan/Causal_genes_GWAS_loci_CAD. Results reported in this paper used version

## Abstract

Multivariate Mendelian randomization (MVMR) is a statistical technique that uses sets of genetic instruments to estimate the direct causal effects of multiple exposures on an outcome of interest. At genomic loci with pleiotropic gene regulatory effects, that is, loci where the same genetic variants are associated to multiple nearby genes, MVMR can potentially be used to predict candidate causal genes. However, consensus in the field dictates that the genetic instruments in MVMR must be independent (not in linkage disequilibrium), which is usually not possible when considering a group of candidate genes from the same locus. Here we used causal inference theory to show that MVMR with correlated instruments satisfies the instrumental set condition. This is a classical result by Brito and Pearl (2002) for structural equation models that guarantees the identifiability of individual causal effects in situations where multiple exposures collectively, but not individually, separate a set of instrumental variables from an outcome variable. Extensive simulations confirmed the validity and usefulness of these theoretical results. Importantly, the causal effect estimates remained unbiased and their variance small even when instruments are highly correlated, while bias introduced by horizontal pleiotropy or LD matrix sampling error was comparable to standard MR. We applied MVMR with correlated instrumental variable sets at genome-wide significant loci for coronary artery disease (CAD) risk using expression Quantitative Trait Loci (eQTL) data from seven vascular and metabolic tissues in the STARNET study. Our method predicts causal genes at twelve loci, each associated with multiple colocated genes in multiple tissues. We confirm causal roles for *PHACTR1* and *ADAMTS7* in arterial tissues, among others. However, the extensive degree of regulatory pleiotropy across tissues and the limited number of causal variants in each locus still require that MVMR is run on a tissue-by-tissue basis, and testing all gene-tissue pairs with *cis*-eQTL associations at a given locus in a single model to predict causal gene-tissue combinations remains infeasible. Our results show that within tissues, MVMR with dependent, as opposed to independent, sets of instrumental variables significantly expands the scope for predicting causal genes in disease risk

1.0.0 of the code, which has been archived at \url{https://doi.org/10.5281/zenodo.10091331}. Supporting data is available at https://dataverse.no/dataset.xhtml?persistentId=doi:10.18710/VM0WKQ.

**Funding:** T.M. acknowledges support from the Research Council of Norway (project number 312045). J.L.M.B. acknowledges support from the Swedish Research Council (2018-02529 and 2022-00734), the Swedish Heart Lung Foundation (2017-0265 and 2020-0207), the Leducq Foundation AteroGen (22CVD04) and PlaqOmics (18CVD02) consortia; the National Institute of Health-National Heart Lung Blood Institute (NIH/NHLBI, R01HL164577; R01HL148167; R01HL148239, R01HL166428, and R01HL168174), American Heart Association Transformational Project Award 19TPA34910021, and from the CMD AMP fNIH program. T.M. and J.L.M.B. acknowledge the European Union's Horizon Europe (European Innovation Council) programme (grant agreement number 101115381). The funders had no role in study design, data collection and analysis, decision to publish, or preparation of the manuscript.

**Competing interests:** The authors have declared that no competing interests exist.

loci with pleiotropic regulatory effects. However, considering risk loci with regulatory pleiotropy that also spans across tissues remains an unsolved problem.

---

## Author summary

Although genome-wide association studies have mapped thousands of genetic variants that explain the heritable nature of many complex traits and diseases, the causal genes and mechanisms underlying these associations are often unclear. This is partly due to the widespread presence of "regulatory pleiotropy", a phenomenon where the same genetic variants affect gene expression of multiple genes in the same genomic locus across multiple tissues. Mendelian randomization is a statistical method that uses genetic variants as instrumental variables to estimate causal effects of exposures on outcomes. Here we have extended this technique to the situation where multiple exposures can have a simultaneous effect on an outcome, and no independent instrumental variables are available for each exposure. When applied to a dataset of genetic and gene expression variation in seven vascular and metabolic tissues of 600 individuals undergoing heart surgery, our method identified candidate causal genes and tissues for coronary artery disease risk at genomic positions where regulatory pleiotropy and the extensive correlations between genetic variants made the application of existing Mendelian randomization methods infeasible. Further support for the validity of our method to identify causal genes using sets of correlated instrumental variables was provided by extensive simulations and theoretical results.

## Introduction

Mendelian randomization (MR) is a statistical technique that uses genetic instruments to estimate causal effects between complex traits and diseases [1–3]. It is based on the fact that genotypes are independently assorted and randomly distributed in a population by Mendel's laws, and not affected by environmental or genetic confounders that affect both traits (commonly called the "exposure" and the "outcome" trait for the putative causal and affected trait, respectively). If it can be assumed that a genetic locus affects the outcome only through the exposure, then the causal effect of the exposure on the outcome can be derived from their relative associations to the genetic locus, which acts as an instrumental variable [4].

Traditionally MR is used to study phenotypic traits where genetic effect sizes are small and horizontal pleiotropy can never be fully excluded, that is, alternative causal paths may exist between the genetic instruments and the outcome trait independent of the exposure trait. Hence much of the methodological development in MR has sought to address these limitations [5, 6].

More recently, MR has been applied to estimate causal effects of *molecular* traits such as mRNA or protein abundances on phenotypic traits [7–10]. Using molecular exposure traits in MR offers potential advantages due to genetic effects on molecular abundances being much larger than those for phenotypic traits, with greater than two-fold allelic effect sizes [11] on gene expression not being uncommon [12]. Moreover, fundamental molecular biological principles reduce the risk of horizontal pleiotropy. Indeed, most genetic variants associated to mRNA and protein abundances (and phenotypic traits) are located in non-coding regions of the genome and there are no known mechanisms by which such variants could affect outcome

traits other than through affecting transcription and translation of nearby genes, typically considered to be at most a distance of 1 Mbp away [13]. Hence, by limiting the selection of genetic instruments to those in the genomic vicinity of the gene whose mRNA or protein product is used as the exposure in an MR analysis, the risk of these variants affecting the outcome through alternative pathways not including the exposure gene or protein is reduced.

However, analyses of large transcriptomic datasets have shown that genetic variants with "regulatory pleiotropy", defined as variants associated with more than one gene in the same locus [12], are common. For instance, it has been found that around 10% of non-redundant cis-expression quantitative trait loci (cis-eQTLs; single nucleotide polymorphisms (SNPs) exhibit regulatory pleiotropy [14]. This covered cases where a true cis-regulatory DNA region was shared between genes as well as cases where genes had distinct regulatory elements that were in high linkage disequilibrium (LD) with one another. More recently, analyses by the GTEx Consortium of RNA abundances across 49 human tissues showed that a median of 57% of variants per tissue are associated with more than one gene in the same locus, typically co-occurring across tissues [12].

When genes or proteins are tested as exposures one-by-one using MR, as in most studies to date, regulatory pleiotropy could clearly lead to horizontal pleiotropy as one cannot know *a priori* to which of the associated genes a genetic instrument exerts its influence on the outcome. Hence, to account for regulatory pleiotropy, all MR analyses using molecular abundances must employ *multivariable* MR (MVMR) methods, where all gene products in a genomic locus of interest are treated simultaneously as a set of potential causes for an outcome of interest.

As with single-exposure MR, MVMR was first developed for phenotypic exposure traits [15–18]. MVMR requires a set of instrumental variable SNPs, at least as many as the number of exposures that are associated with the exposure and outcome variables, and not affecting the outcome other than through the set of exposure variables. MVMR can then estimate the direct causal effect of each exposure on the outcome using a two-stage least squares method in the single-sample, individual-level data setting, or a "regression of regression coefficients" method in the two-sample summary data setting [15–17]. Recently, MVMR was applied to transcriptomic and genome-wide association study (GWAS) data in the two-sample setting, in a first attempt to overcome the challenge of regulatory pleiotropy when analyzing mRNA abundances to identify causal disease genes, in a procedure called transcriptome-wide MR (TWMR) [19].

However, the precise conditions to ensure validity of MVMR remain somewhat ambiguous, in particular with regards to the presence of linkage disequilibrium (correlations) between the SNPs in the set of instrumental variables. For instance, Burgess *et al.* [16] state that estimates from the two-stage least squares method in the individual-level data setting are valid even if the genetic variants are in LD [15], but require uncorrelated instruments in their proof of equivalence with the regression method in the summary data setting. McDaid *et al.* [20] show that the two-stage least squares estimates can be obtained from univariate summary data even if the genetic variants are in LD, allowing for two-sample MVMR analysis, albeit at the cost of the estimates no longer being expressed as a "regression of regression coefficients". Nevertheless, despite their formula for the causal effect estimates being valid for correlated instruments, McDaid *et al.* [20] still follow common practice in MVMR to select only instruments spread far apart on the genome to eliminate any LD between them. Similarly, Porcu *et al.* [19], whose TWMR method is based on the analytic results of McDaid *et al.* [20], only use variants with low mutual LD ($r^2 \leq 0.1$) in their analysis of loci with regulatory pleiotropy. Moreover, all simulations reported in the literature to test MVMR methods in a controlled situation use independently sampled instruments [15–17, 19].

Here we clarify and strengthen the theoretical underpinnings of MVMR with correlated instrumental variables, supported by realistic simulations, to identify causal genes at genomic loci with pleiotropic gene regulatory effects. If we restrict to genetic instruments in the same locus to exclude horizontal pleiotropy, then inclusion of correlated instruments becomes a necessity.

The basis of our approach is Wright's method of path coefficients [21, 22] and its generalization into the graph structural causal inference theory by Pearl and colleagues [23]. Accordingly, we distinguish between *identification* and *estimation* of causal effects. Causal identification refers to the question whether causal effects can be expressed in terms of the true (but unknown) joint distribution of the variables in a causal graph. Causal estimation refers to the problem of estimating the identified causal effects from a finite number of observational samples from the joint distribution.

For the identification problem, we show that the MVMR causal graph with correlated instruments belongs to a class of causal graphs previously studied in the AI literature [24]. Specifically, we show that any set of non-redundant causal variants of the same size as the set of exposure traits, regardless of their mutual LD levels and regardless of any interactions between the exposures, satisfies Brito and Pearl's [24] *"instrumental set"* condition, allowing identification of the causal effects of each exposure on the outcome. In particular, the causal effects can be expressed as the coefficients of the linear regression of the instrument-outcome covariances on the instrument-exposure covariances, thereby showing correctness of the "regression of regression coefficients" method in all settings.

For the estimation problem, we expand the instrument set to the overcomplete set of all genetic variants in the locus of interest. We assume that this set contains at least as many causal variants as the number of exposures, and analyze a class of methods called the Generalized Method of Moments (GMM) [25]. As shown by Hansen [25], every previously suggested instrumental variables estimator, in linear or nonlinear models, with cross-section, time series, or panel data, as well as system estimation of linear equations can be cast as a GMM estimator. GMM is therefore viewed as a unifying framework for estimation in causal inference problems.

We show that for the exactly determined MVMR problem with correlated instruments (equal number of instruments and exposures), replacing the true covariances by their finite-sample estimates is the optimal GMM estimator. For the overdetermined system, a weighted version that corresponds exactly to the two-stage least squares solution is optimal. In particular, the solution of McDaid *et al.* [20] of the two-stage least squares problem in terms of summary statistics is shown to be the finite-sample GMM estimator of the "regression of regression coefficients" solution to the causal identification problem.

To support our theoretical results, we conducted extensive simulations with binomially distributed, correlated instruments mimicking real LD from the human genome. We further applied our MVMR method with correlated instrumental variable sets to transcriptomics data from seven vascular and metabolic tissues in 600 coronary artery disease (CAD) cases undergoing surgery from the STARNET study [26] to predict causal genes and tissues at 19 genome-wide significant CAD risk loci with regulatory pleiotropy.

## Materials and methods

### The method of path coefficients

Causal effect identification consists of expressing the effect of an intervention of one variable on the marginal distribution of one or more other variables using knowledge of the underlying causal diagram and joint distribution of all variables under consideration [23]. A causal

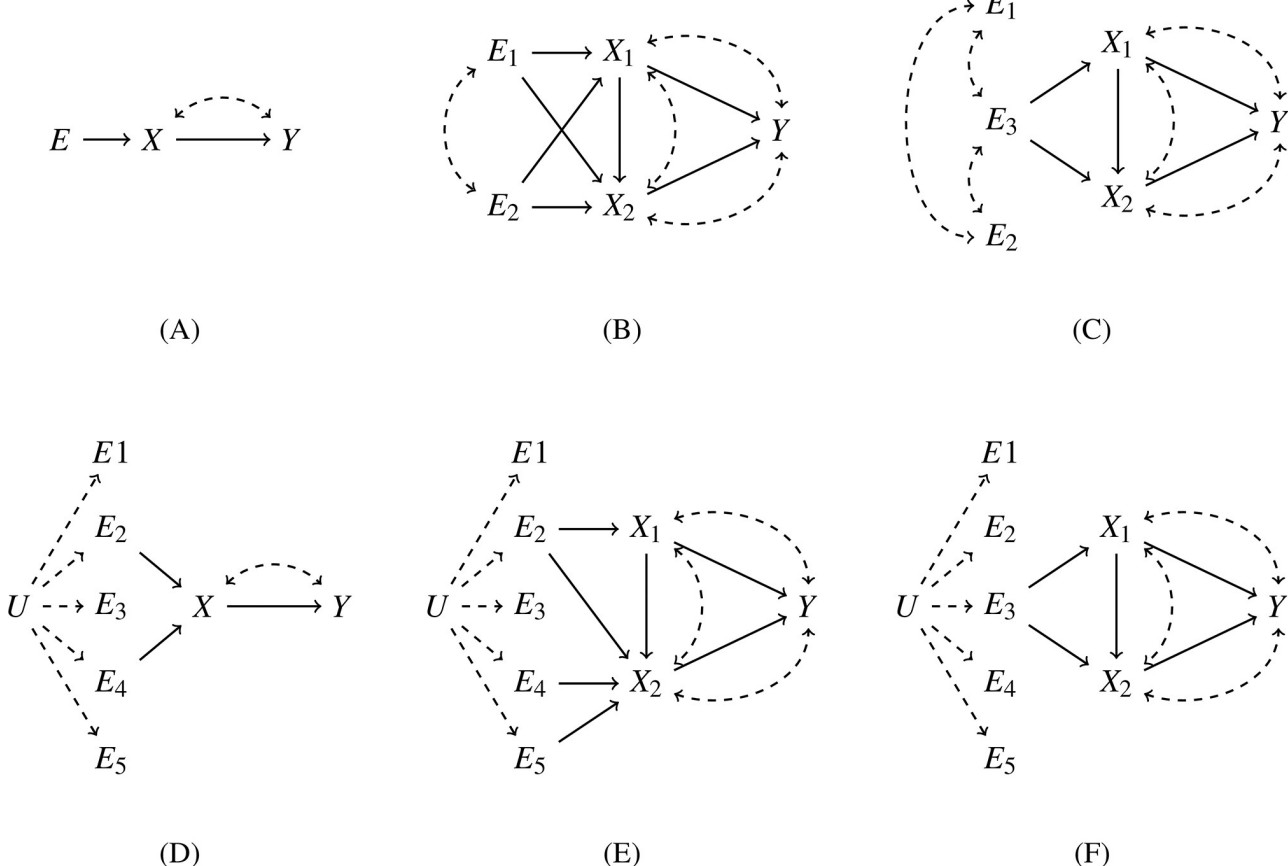

**Fig 1. Causal diagrams for identification (A-C) and estimation (D-F) of causal effects at GWAS loci with regulatory pleiotropy. (A)** The standard instrumental variable graph is only suitable for modelling the causal effect of gene $X$ on outcome trait $Y$ if $X$ is the *only* cis-eGene in the locus of the genome-wide significant variant $E$ for $Y$. **(B)** A causal graph for MVMR with regulatory pleiotropy, where two or more cis-eGenes $X_i$ are found in a genome-wide significant locus for $Y$; the causal effects of each $X_i$ on $Y$ can be identified if there exist at least the same number of causal variants as the number of cis-eGenes in the GWAS locus, even when they are in LD with each other, because the variants form an *instrumental set*. **(C)** If the number of causal variants is smaller than the number of cis-eGenes and other variants in the locus are merely associated by LD, the causal effects of $X_i$ on $Y$ are non-identifiable. **(D,E)** When estimating causal effects from finite samples, we use an overdetermined set of genetic variants as instruments, without knowing the number or identity of the causal variants. **(F)** If the (unknown) number of causal variants in a locus is smaller than the number of cis-eGenes, causal effects can not be estimated, a condition that can be tested by analyzing the rank or determinant of the covariance matrix between the variants and cis-eGenes. Solid arrows represent direct causal effects and dashed (bi)directed arrows represent non-zero covariances due to unobserved ($U$) confounders.

diagram is a directed acyclic graph (DAG) of directed edges between variables where a causal relationship is known or hypothesized, augmented with bidirected edges between variables known or hypothesized to be affected by unmodelled confounding factors (Fig 1); bidirected edges can be represented equivalently by two directed edges originating from a common latent factor.

A special class of causal models are those where the causal diagram is assumed to define a linear structural equation model (SEM) [27], that is, a system of linear equations among a set of variables $\{X_1, \ldots, X_p\}$ of the form

$$X_i = \sum_j c_{ij} X_j + U_i.$$

The $U_i$ are error or disturbance terms representing unobserved background factors which are assumed to have mean zero and a variance/covariance matrix $[\Psi_{ij}] = \text{Cov}(U_i, U_j)$. It is

often assumed that the error terms are normally distributed, but, as already pointed out by Wright [22], this is not necessary—the results below hold for any error distribution. Every non-zero coefficient $c_{ij}$ corresponds to a directed edge $X_j \rightarrow X_i$ in the causal diagram, that is, $c_{ij}$ is the direct causal effect of $X_j$ on $X_i$, and every non-zero element $\Psi_{ij}$ corresponds to a bidirected edge $X_i \leftrightarrow X_j$. The SEM structure implies that the variables $\{X_1, \ldots, X_p\}$ have mean zero and covariance matrix

$$\Sigma = (I - C)^{-1} \Psi (I - C^T)^{-1}$$

where $C$ is the matrix of coefficients $c_{ij}$, $C^T$ is its transpose, and $I$ is the identity matrix. Causal identification in linear SEMs consists of solving for the non-zero $c_{ij}$ (and $\Psi_{ij}$) in terms of the covariance matrix $\Sigma$.

The method of path coefficients is a result derived by Sewall Wright [21, 22, 27], which expresses the covariance between any pair of variables in a linear SEM as a polynomial in the parameters (causal coefficients and error covariances) of the model:

$$\sigma_{XY} = \sum_p T(p) \tag{1}$$

where the summation ranges over unblocked paths $p$ connecting $X$ and $Y$, and $T(p)$ is the product of the parameters ($c_{ij}$ or $\Psi_{ij}$) of the edges along $p$. A path in a graph is a sequence of edges such that each pair of consecutive edges share a common node and each node appears only once along the path. A path is unblocked if it does not contain a collider, that is, a pair of consecutive edges pointing at the common node. Two nodes are called $d$-separated if there are no unblocked paths between them. Eq (1) is valid for standardized variables that are normalized to have zero mean and unit standard deviation; if this is not the case, a modified rule can be applied where each $T(p)$ is multiplied by the variance of the variable that acts as the "root" for path $p$ [27].

## Instrumental sets

Fix a variable $Y$ in a linear SEM, and assume we have the equation

$$Y = c_1 X_1 + \cdots + c_K X_K + U_Y$$

in our SEM. Brito and Pearl [24] derived a sufficient graphical condition for identifying the parameters $c_1, \ldots, c_K$ by application of Wright's rule (1), of which we present a simplified version. The set of variables $\{E_1, \ldots, E_K\}$ is said to be an "instrumental set" relative to $\{X_1, \ldots, X_K\}$ if there exist pairs $(E_1, p_1), \ldots, (E_K, p_K)$ such that for each $i$ the following three conditions hold.

1. $E_i$ is a non-descendant of $Y$ and $p_i$ is an unblocked path between $E_i$ and $Y$ including edge $X_i \rightarrow Y$.

2. $E_i$ is $d$-separated from $Y$ in $\bar{\mathcal{G}}$, the graph obtained after removing the edges $X_1 \rightarrow Y, \ldots, X_k \rightarrow Y$ from the original graph $\mathcal{G}$.

3. For $j > i$, the variable $E_j$ does not appear in path $p_i$, and if paths $p_i$ and $p_j$ have a common variable $V$, then both $p_i[V \sim Y]$ and $p_j[E_j \sim V]$ point to $V$, where $p_i[V \sim Y]$ and $p_j[E_j \sim V]$ denote truncations of $p_i$ and $p_j$ from $V$ and until $V$, respectively.

If $\{E_1, \ldots, E_K\}$ is an instrumental set relative to $\{X_1, \ldots, X_K\}$, then the parameters of the edges $X_1 \rightarrow Y, \ldots, X_K \rightarrow Y$ are identifiable and can be computed by a set of linear equations [24]. For concrete examples verifying the instrumental set conditions, see S1 File.

## Causal effect identification in MVMR with correlated instruments

We assume genes $X_1, \ldots, X_K$ are colocated in the same genomic locus and have a potential causal effect on a phenotypic trait $Y$ associated to the locus. We assume there exist $K$ causal variants in the locus such that each gene has at least one causal variant (causal variants may be shared between genes). The variants are correlated by LD, the exposures $X_i$ can be connected by causal relations or unmodelled latent factors, and likewise there can be latent factors affecting the exposures $X_i$ and outcome $Y$. Randomization of genotypes ensures that there are no unmodelled correlations between the $E_j$ and $X_i$. An example causal diagram for $K = 2$ is shown in Fig 1B.

We now show that $\{E_1, \ldots, E_K\}$ is an instrumental set relative to $\{X_1, \ldots, X_K\}$. By relabelling, we may assume that there exists an edge $E_i \rightarrow X_i$ in the causal diagram. Define the paths $p_i = \{E_i \rightarrow X_i, X_i \rightarrow Y\}$. Since $E_i$ is $d$-separated from $Y$ in the diagram where *all* edges $X_j \rightarrow Y$ are removed, it follows that the set $\{(E_1, p_1), \ldots, (E_K, p_K)\}$ satisfies the three instrumental set conditions (see above) and the causal parameters $c_i$ are identifiable. We apply Wright's rule (Eq (1)), following the general proof of Brito and Pearl [24]. By Eq (1),

$$\sigma_{E_i Y} = \sum_p T(p) \tag{2}$$

where the sum is over unblocked paths $p$. It is clear that any path that starts in $E_i$, ends in $Y$, and includes a bidirected edge $X_j \leftrightarrow Y$ must be blocked by a collider $V \rightarrow X_j \leftrightarrow Y$, where $V$ can be either a variant or another gene. Hence no such paths enter the sum in Eq (2). In other words, all unblocked paths must end with an edge $X_j \rightarrow Y$ contributing a factor $c_j$ to $T(p)$. Collecting all paths by their final edge, the sum can be decomposed as

$$\sigma_{E_i Y} = \sum_j c_j \sum_{p': E_i \sim X_j} T(p')$$

where the inner sum now extends over the subset of paths from $E_i$ to $X_j$ obtained by truncating the unblocked paths from $E_i$ to $Y$ which have $X_j$ as their penultimate node. Since the truncation of an unblocked path is also necessarily unblocked, each $p'$ is an unblocked path from $E_i$ to $X_j$. Vice versa, every unblocked path from $E_i$ to $X_j$ can be extended to an unblocked paths from $E_i$ to $Y$ by adding the edge $X_j \rightarrow Y$. Hence, the sum over $p'$ is the sum over *all* unblocked paths from $E_i$ to $X_j$, By another application of Wright's rule (1),

$$\sum_{p': E_i \sim X_j} T(p') = \sigma_{E_i X_j},$$

and hence

$$\sigma_{E_i Y} = \sum_j c_j \sigma_{E_i X_j}.$$

In matrix-vector notation, we obtain:

$$\Sigma_{EX} c = \Sigma_{EY},$$

where $c = (c_1, \ldots, c_K)^T$ is the vector of causal effects, and $\Sigma_{EX}$ and $\Sigma_{EY}$ are the matrix and vector of covariances $[\sigma_{E_i X_j}]$ and $(\sigma_{E_1 Y}, \ldots, \sigma_{E_K Y})^T$, respectively.

Since $\Sigma_{EX}$ is not a symmetric matrix, we cannot assume that an inverse exists, but we can always write the solution to the linear set of equations using a generalized left inverse:

$$c = (\Sigma_{EX}^T \Sigma_{EX})^{-1} \Sigma_{EX}^T \Sigma_{EY}, \tag{3}$$

that is, find the least squares solution. These equations are valid for standardized variables.

## The generalized method of moments for causal effect estimation in MVMR with correlated instruments

To estimate the causal effects from observational data, we consider multiple finite-sample approximations to the above equations. For finite-sample estimation, the number of instruments can be greater than the number of exposures, that is, we assume we have $N$ observations of the random variables $\{E_1, \ldots, E_L\}$, $\{X_1, \ldots, X_K\}$, and $Y$, with $L \geq K$. We use lower-case letters to denote sample values, e.g. $e_{il}$ denotes the value of $E_l$ in observation $i$. We use the notation $e_{i\cdot}$, $e_{\cdot l}$, and $e_{\cdot\cdot}$ to represent the vector of sample values for all $L$ instruments in observation $i$, the vector of all $N$ observations of instrument $l$, and the $N \times L$ matrix of all observations for all instruments, respectively, and similarly for $x_{i\cdot}$, $x_{\cdot k}$, and $x_{\cdot\cdot}$. We assume the observations for all variables are standardized to mean zero and unit standard deviation.

The generalized method of moments (GMM) [25] is based on moment functions that depend on observable random variables and unknown parameters, and that have zero expectation in the population when evaluated at the true parameters:

$$\mathbb{E}[g(w_i, c)] = 0 \tag{4}$$

where $g$ denotes the moment function, $w_i$ is a vector of samples of random variables in observation $i$, and $c$ is a vector of unknown parameters. The moment function $g$ can be linear or nonlinear.

Assuming a linear MVMR SEM as before, the natural moment functions are of the form

$$g([e_{il}, x_{i\cdot}, y_i], c) = e_{il}(y_i - x_{i\cdot}^T c), \tag{5}$$

that is, we consider one moment equation per instrument. From the structural equation for $Y$ in the assumed SEM (see above and Fig 1), it follows that $Y - X^T C = U_Y$, the residual error on $Y$, and since all instruments $E_l$ are assumed independent of $U_Y$ (no bidirected arrows between $E_l$ and $Y$), that is, $\mathrm{Cov}(E_l, U_Y) = \mathbb{E}(E_l U_Y) = 0$, it follows immediately that the moment conditions are satisfied:

$$\mathbb{E}[e_{il}(y_i - x_{i\cdot}^T c)] = 0$$

Replacing the expectation $\mathbb{E}$ with the empirical mean over the observations results in $L$ equations in $K$ unknowns $(c_1, \ldots, c_K)$ which can be written in matrix-vector notation as

$$\frac{1}{N}\sum_{i=1}^{N} e_{il}(y_i - x_{i\cdot}^T c) = e_{\cdot l}^T(y - x_{\cdot\cdot}c) = 0 \tag{6}$$

If the number of instruments equals the number of unknowns, $L = K$, this system can be solved exactly, and by the standardization of all variables, the solution reduces to the least-squares estimator $\hat{c}_{LS}$,

$$\hat{c}_{LS} = (\hat{\Sigma}_{EX}^T \hat{\Sigma}_{EX})^{-1} \hat{\Sigma}_{EX}^T \hat{\Sigma}_{EY}, \tag{7}$$

which is also obtained by replacing the true covariances in the instrumental variable formula (3) by their empirical estimates.

If the model is overdetermined, $L > K$, then in general there is no unique solution to (6), because not all $L$ sample moments will hold exactly. While the least-squares estimator provides one approximate solution to the overdetermined system, Hansen [25] proposed an alternative solution, based on bringing the sample moments as close to zero as possible by minimizing the quadratic form

$$\left[\frac{1}{N}\sum_{i=1}^{N} g([e_{il}, x_{i.}, y_i], c)\right]^T \cdot \Delta \cdot \left[\frac{1}{N}\sum_{i=1}^{N} g([e_{il}, x_{i.}, y_i], c)\right] \quad (8)$$

with respect to the parameters $c$, where $\Delta$ is a positive definite $L \times L$ weighting matrix, and the quantity between the large square brackets is understood to be the $L$-vector of moment functions. It can be shown that this yields a consistent estimator $\hat{c}_{GMM}(\Delta)$ of $c$, under certain regularity conditions. Again using the standardization of all variables, the cost function can be written as

$$(\hat{\Sigma}_{EY} - \hat{\Sigma}_{EX}c)^T \cdot \Delta \cdot (\hat{\Sigma}_{EY} - \hat{\Sigma}_{EX}c),$$

with solution

$$\hat{c}_{GMM}(\Delta) = (\hat{\Sigma}_{EX}^T \Delta \hat{\Sigma}_{EX})^{-1} \hat{\Sigma}_{EX}^T \Delta \hat{\Sigma}_{EY} \quad (9)$$

When $\Delta = I$, the identity matrix, we again obtain the least-squares estimator, $\hat{c}_{GMM}(I) = \hat{c}_{LS}$.

While all choices of $\Delta$ lead to a consistent estimator, in finite samples different $\Delta$-matrices will lead to different point estimates. Hansen [28] showed that the GMM estimator with the smallest asymptotic variance is obtained by taking $\Delta$ equal to the inverse of the variance-covariance matrix of the moment functions (5). If the errors are homoskedastic, $\mathrm{Var}(U_{y_i}) = \mathrm{Var}(y_i - x_{i.}^T c) = \sigma^2$, we have

$$\Delta = \sigma^{-2}\hat{\Sigma}_{EE}^{-1},$$

and obtain the estimator

$$\hat{c}_{GMM} = (\hat{\Sigma}_{EX}^T \hat{\Sigma}_{EE}^{-1} \hat{\Sigma}_{EX})^{-1} \hat{\Sigma}_{EX}^T \hat{\Sigma}_{EE}^{-1} \hat{\Sigma}_{EY}. \quad (10)$$

We will refer to this estimator as the GMM estimator. Note that this estimator is equivalent to the two-stage least squares estimator.

## Simulations with randomly generated instrument correlations

We simulated data from idealized models corresponding to the causal diagrams in Fig 1 with binomially distributed instruments and normally distributed exposure and outcome variables.

For the diagrams with a single exposure (Fig 1A and 1D), we set the true causal effect $c_X = 0.3$. In the case of one instrument (Fig 1A) we varied the instrument strength $a$ between 0.01 (weak instrument), 0.3 (mediocre instrument), and 0.8 (strong instrument). In the case of multiple instruments (Fig 1D), we simulated ten instruments with randomly sampled instrument strengths, either all strong (effect sizes between $0.1 - 0.3$) or all weak (effect sizes between $0.001 - 0.03$).

For the diagrams with multiple exposures and correlated instruments (Fig 1B, 1C, 1E and 1F), we set the true causal effects $c_{X_1} = 0.2$ and $c_{X_2} = 0.6$ in all simulations.

To test the effect of instrument correlation we simulated data with two instruments (Fig 1B) where the pairwise correlation of the instruments varied between 0.01, 0.3, 0.7, 0.9 and 0.95. The matrix $A = [a_{ij}]$ of instrument effect sizes, where $a_{ij}$ is the direct causal effect of $E_i$ on

$X_j$, was generated randomly such that its determinant was greater than 0.05 and there were no weak instruments ($a_{ij}$ between $0.1 - 0.3$).

To test the effect of weak instruments, we simulated data with two instruments (Fig 1B) with randomly generated non-zero pairwise correlation of the instruments and randomly generated matrix of instrument effect sizes $A$ with determinant either less than 0.001 and effect size of each instrument less than 0.001 or determinant greater than 0.05 and effect size of each instrument between $0.1 - 0.3$.

To test robustness of the causal effect estimates, we simulated data with two instruments and two exposures (Fig 1B), with randomly generated non-zero pairwise correlation between the instruments, randomly generated matrix of instrument effect sizes $A$ with determinant greater than 0.05, and true causal effects of the exposures on the outcome equal to $c_{X_1} = 0.2$ and $c_{X_2} = 0.6$, but estimated causal effects for each exposure separately under the (false) hypothesis that the data was generated from Fig 1A.

## Simulations with instrument correlations derived from real LD matrices

For more realistic simulations, we simulated data with binomially distributed genotypes and normally distributed exposure and outcome variables in such a way that the summary statistics from the simulated data match summary statistics at real GWAS loci for coronary artery disease (CAD). We used eQTL summary statistics from GTEx [12] and CAD GWAS summary summary statistics from the CARDIoGRAMplusC4D meta-analysis [29].

For a given locus of interest with cis-eQTLs $E_1, \ldots, E_L$ for genes $X_1, \ldots, X_K$ ($L \geq K$), with LD matrix $\Sigma_E$ (size $L \times L$), matrix of eQTL summary statistics $A$ (size $L \times K$), and vector of CAD GWAS summary statistics $\Sigma_{EY}$ (size $L \times 1$), we simulated data as follows.

To sample binomially distributed genetic instruments according to a predefined LD matrix, we made the simplifying assumption that the correlation structure between the simulated instruments can be described by a Markov chain. We sampled the first SNP's genotype from a binomial distribution with probability of success equal to its real minor allele frequency (MAF). For every other SNP, we sampled a genotype from a binomial distribution with probability of success conditional on the genotype of the previous SNP in the sequence, such that both the MAF and pairwise LD between successive SNPs match the real data from the simulated locus. Further details are given in S1 File.

We assumed that a randomly selected subset of at least $K$ eQTLs were causal and all others were associated with the exposures only through LD. For the causal eQTLs, we sampled the vector **a** of instrument effect sizes randomly between 0.1 and 0.3,

$$X_j \mid E_1, \ldots, E_K \sim \mathcal{N}(\sum_i a_i E_i, \sigma^2),$$

with $\sigma^2 = 1$.

Finally, we set the causal effect $c$ of $X$ on the outcome $Y$ equal to the value obtained from eq. (3) with the real $\Sigma_{EY}$, and sampled outcome data from

$$Y \mid X \sim \mathcal{N}(cX, \sigma^2),$$

with $\sigma^2 = 1$.

We performed simulations to analyze the influence of LD matrix accuracy, weak instrument bias, standard error calculation method, and horizontal pleiotropy.

For the simulations on the accuracy of the LD matrix, we generated datasets for the *SLC22A3-LPA-PLG* locus (centered on chr 6: 161089307) in liver as exposures. Causal effects of the genes on the outcome variable $Y$ were calculated to be $c_{SLC22A3} = 0.15$, $c_{LPA} = -0.05$ and

$c_{PLG}$ = −0.27 when seven instruments were used, and the same values were also used in the other simulations. We compared the situations where $L$ = 3 (LD threshold of 0.25), $L$ = 4 (LD threshold of 0.3), $L$ = 5 (LD threshold of 0.4), $L$ = 6 (LD threshold of 0.5), and $L$ = 7 instruments (LD threshold of 0.8) were used. We performed one-sample simulations where the sample size varied between 500 − 30000. For the instrument effect sizes, we assumed that three eQTLs were causal for all genes (their effect sizes sampled from a uniform distribution within the range 0.1 − 0.3), while additional eQTLs were assumed to be associated with the exposures only through LD.

For the simulations on weak instrument bias, we generated datasets within the same locus and setup as the LD accuracy simulations except here we kept $L$ = 8 eQTLs (LD threshold of 0.96) and varied the sample size between 500 − 10000. For the instrument effect sizes, we again assumed that only three eQTLs were causal for all genes, their effect sizes sampled from a uniform distribution within the range 0.1 − 0.3 for strong instruments and within the range 0.001 − 0.01 for weak instruments. We confirmed that these instruments were indeed weak/strong by employing the *conditional F-statistics* computed using the *Mendelian Randomization* package [30] as well as the *MVMR* package [31] (S3 Fig).

For the simulations on comparing the standard errors computed approximately using summary-level data and exactly using individual-level data, we generated datasets within the same locus and setup as the previous simulations. We kept $L$ = 7 eQTLs (LD threshold of 0.95) and varied the sample size between 500 − 10000. For the instrument effect sizes, we again assumed that only three eQTLs were causal for all genes. Details about the standard error calculation using summary and individual-level data are given in detail in S1 File.

To simulate the effect of horizontal pleiotropy, we generated datasets for the *SLC22A3-LPA-PLG* locus with causal effects of the genes on the outcome variable $Y$ set to $c_{LPA}$ = −0.05 and $c_{PLG}$ = −0.27 and $c_{SLC22A3} \in [0.0, 0.4]$. We used eight instruments where only three eQTLs were causal for all genes, their effect sizes sampled from a uniform distribution within the range 0.1 − 0.3. We estimated the effects in a misspecified estimation model where we performed MVMR only using *LPA-PLG* genes, assuming horizontal pleiotropy through one unknown/unmeasured gene *SLC22A3* in the same locus.

To simulate the effect of potential bias introduced by two-sample MR, we generated independently sampled eQTL and GWAS datasets for the *MRAS-ESYT3* locus in the Aorta tissue of STARNET, which share $L$ = 5 eQTLs in their locus (centred on chr 3:138121920), with causal effects of the genes on the outcome variable $Y$ set to $c_{MRAS}$ = 0.208 and $c_{ESYT3}$ = −0.294. We simulated eQTL sample sizes between 300 − 4000 and GWAS sample sizes between 10000 − 140000.

## Comparison to other MVMR methods

We compared the least squares (LS) (Eq (7)) and GMM (Eq (10)) estimators to seven other multi-variable MR methods: Transcriptome-Wide MR (TWMR) [19], the *mv_multiple* estimator from *TwoSampleMR* package [32], and the *mv_mvivw* estimator from the *Mendelian Randomization* package [30], estimators MVMR-Robust, MVMR-Median, and MVMR-Lasso [33] and estimator MVMR-cML [34].

The least squares and GMM estimators can also be applied in the univariate MR setting. In this setting we performed a comparison to all available methods in the MR-Base package, namely, Inverse Variance weighted (IVW), Weigted Median (WM), Maximum Likelihood (MaxLik), MR-Egger (Egger). We also included TWMR in the comparison.

In the course of our analysis we identified the presence of a "regularization" term in the TWMR source code that was hard-coded to a specific value, presumably related to the sample

size of the use-case from their paper [10]. More precisely, for the weight matrix used in the GMM estimator, $H = ((\mathbf{X}^T\mathbf{E}) \cdot (\mathbf{E}^T\mathbf{E})^{-1} \cdot (\mathbf{E}^T\mathbf{X}))^{-1}$, they used shrinkage of the form

$$H_{shrunk} = (1 - \alpha)H + \alpha I$$

Here $\alpha$ is $\frac{1}{\sqrt{N}}$ where $N = 3781$ and $I$ is the identity matrix. Results reported here used the published version of TWMR including this hard-coded term.

## Prediction of causal genes and tissues at GWAS loci for coronary artery disease

We used eQTL summary statistics from the Stockholm-Tartu Atherosclerosis Reverse Network Engineering Task (STARNET) study, a genetics-of-gene expression study of tissue samples from blood ($n = 481$), atherosclerotic-lesion-free internal mammary artery (MAM, $n = 552$), atherosclerotic aortic root (AOR, $n = 539$), subcutaneous fat (SF, $n = 573$), visceral abdominal fat (VAF, $n = 531$), skeletal muscle (SKLM, $n = 534$), and liver (LIV, $n = 576$) obtained during open thorax surgery of 600 coronary artery disease (CAD) patients [26]. eQTL summary statistics from matching tissues in GTEx [12] were used for replication analyses. Significant cis-eQTL associations were defined as in the original STARNET study (FDR <5% across all tested SNP-gene combinations where the SNP is within 1Mb up or downstream of the center of the gene).

We used coronary artery disease (CAD) as the outcome trait and used summary statistics from the CARDIoGRAMplusC4D genome-wide association meta-analysis (GWAMA) study [29]. Summary statistics specific to the European population were extracted using the TwoSampleMR package [32] (study IDs ebi-a-GCST003116, $n = 141, 217$).

To define locus boundaries and candidate genes of interest for MVMR, we considered genome-wide significant SNPs, all their linked eQTLs within a $0.5Mb$ radius, and all the genes they are associated with in one or more tissues ("eGenes"). Specifically, we first created tissue-specific outcome data with GWAS summary statistics for all SNPs with non-zero (FDR <5%) eQTL effect in the STARNET summary statistics for the tissue of interest. We filtered these eQTLs by their GWAS p-values, retaining only those with p-values below $5 \times 10^{-8}$ in the GWAS study. We then iterated through these eQTLs, sorted by GWAS p-value, such that for the first iteration, the first eQTL would be the first lead SNP. We collected all eQTLs within a $0.5Mb$ radius and with non-zero LD with the lead SNP, and then removed the lead SNP as well as its collection of linked eQTLs from the list, repeating the procedure with the next lead SNP. For each lead SNP and its collection of linked eQTLs, we first removed SNPs in perfect LD with the lead SNP and then treated the remaining eQTLs as instrumental variables in one causal diagram containing all genes associated with these eQTLs either in a specific tissue, or across all tissues, depending on the application, to complete the diagram. This procedure ensures that for each locus of interest, we obtain a a closed causal diagram, where "closed" means that no more genes (exposures) can be added to the diagram that have shared cis-eQTL associations with any eQTLs (instruments) either already in the diagram or linked to eQTLs in the diagram.

We computed *conditional F-statistics* using the *Mendelian Randomization* [30] and *MVMR* [31] packages to verify instrument strength.

In replication analyses using GTEx, the same procedure was implemented independently for the same GWAS loci in matching tissues, using *all cis*-eQTL associations in GTEx (which may differ from the *cis*-eQTL associations in STARNET in the same locus). Hence, replications in GTEx represent the best possible causal effect estimation in GTEx and are not biased by the exposure gene sets used in STARNET.

Since in the GWAS study, the log-odds ratio was used as the unit, that is, *logistic* regression was performed of the dichotomous outcome trait on SNP genotype, the $\beta$-value for a SNP is the increase in log-odds of the outcome. Hence in our models, the predicted causal effect of a gene $X$ on the outcome $Y$ (CAD) is to be interpreted as the effect of $X$ on the log-odds ratio of $Y$.

We computed summary-based standard errors using the sample size from the GWAS study (see S1 File for details) and obtained p-values from its asymptotic normal distribution. We confirmed using simulations that summary-based and individual-level based standard errors were comparable (S6 Fig). We further confirmed standard errors using the "mr_mvivw" method of the "MendelianRandomization" package (S6 Fig). We note that to the best of our knowledge, the only standard error estimator that takes into account variability due to the different sample sizes in the two-sample setting [35] requires individual-level data and can thus not be computed in the two-sample summary-data setting.

## Results

### Causal effect identification and estimation in MVMR with correlated instruments

To analyze the problem of MVMR with correlated instruments, we consider causal diagrams where one or more genes $X_i$ are colocated in the same genomic locus and have a potential causal effect on a phenotypic trait $Y$ associated to the same locus. We assume that there exist at least as many causal variants in the locus as there are potential causal genes, such that each gene has at least one causal variant (causal variants may be shared between genes).

If there is only one gene in the locus, we have the classical instrumental variable graph that underpins standard (univariate) MR (Fig 1A). Application of the method of path coefficients (see Methods) shows immediately that the causal effect $c$ of the exposure gene $X$ on the outcome $Y$ is given by the usual instrumental variable formula $c = \sigma_{EY}/\sigma_{EX}$, where $E$ is a causal variant (or a variant in LD with a causal one) for $X$ and $\sigma_{EY}$ and $\sigma_{EX}$ are the covariances between $E$ and $X$, and $E$ and $Y$, respectively.

In the case of regulatory pleiotropy, there will be multiple putative causal genes in the same locus, and we obtain an MVMR causal graph with correlated instruments (Fig 1B). Using analytical results derived by Brito and Pearl [24], it can be shown that the variants $\{E_1, \ldots, E_K\}$ in this graph form an *instrumental set* relative to the exposures (genes) $\{X_1, \ldots, X_K\}$ (see Methods), which permits identification of the direct causal effects $c = (c_1, \ldots, c_K)^T$ associated to the edges $X_1 \to Y, \ldots, X_K \to Y$ in the underlying linear structural equation model (SEM). Application of the method of path coefficients shows that the vector $c$ of causal effects satisfies the system of equations (see Methods for details)

$$\Sigma_{EX}c = \Sigma_{EY}, \tag{11}$$

with solution

$$c = (\Sigma_{EX}^T \Sigma_{EX})^{-1} \Sigma_{EX}^T \Sigma_{EY}, \tag{12}$$

where $\Sigma_{EX}$ and $\Sigma_{EY}$ are the matrix and vector of covariances $[\sigma_{E_iX_j}]$ and $(\sigma_{E_1Y}, \ldots, \sigma_{E_KY})^T$, respectively. We note the similarity between Eqs (11) and (12) and the standard instrumental variable solution for univariate MR.

Eqs (11) and (12) describe the causal effects as the coefficients in a linear relation between univariate covariances, and this relation is *exact* in the infinite sample limit, regardless of LD levels between the instruments. Furthermore, Eqs (11) and (12) remain valid if the number of

genetic variants is greater than the number of candidate genes, regardless whether the additional variants are causal or not (Fig 1D and 1E)—the linear system is then merely overdetermined. However, if the number of true causal variants is less than the number of exposures (Fig 1C and 1F), adding genetic instruments that are only in LD with the causal variants does not help. In this case, the variants violate the third instrumental set condition (see Methods), and Eq (11) is easily seen to be *underdetermined*.

The covariances in Eqs (11) and (12) are the true covariances of the random variables $E_i$, $X_j$, and $Y$. To estimate the causal effects from observational data, we need finite-sample approximations to these equations. The most straightforward estimates $\hat{c}$ are obtained by replacing the true covariances by their empirical estimates. We refer to this solution as $\hat{c}_{LS}$, the least squares (LS) estimator, that is,

$$\hat{c}_{LS} = (\hat{\Sigma}_{EX}^T \hat{\Sigma}_{EX})^{-1} \hat{\Sigma}_{EX}^T \hat{\Sigma}_{EY}.$$

Since the empirical covariances can be obtained from univariate regressions, the least-squares estimator is also called the "regression of regression coefficients" method.

The least-squares estimator belongs to a more general family of estimators obtained by the generalized method of moments (GMM; see Methods for details), which can be written as

$$\hat{c}_{GMM}(\Delta) = (\hat{\Sigma}_{EX}^T \Delta \, \hat{\Sigma}_{EX})^{-1} \hat{\Sigma}_{EX}^T \Delta \, \hat{\Sigma}_{EY}$$

where $\Delta$ is any square invertible matrix with dimensions equal to the number of instrumental variables. If $\Delta$ is the identity matrix (or if $\hat{\Sigma}_{EX}$ is invertible), we recover the least-squares estimator. The estimator $\hat{c}_{GMM}(\Delta)$ is consistent for all choices of $\Delta$. With homoskedastic errors, the estimator with theoretically minimal variance is obtained by taking $\Delta = \hat{\Sigma}_{EE}^{-1}$, the inverse of the empirical covariance matrix of the instruments, that is, the inverse of the LD matrix of the variants (see Methods for details). We will refer to this estimator as the GMM estimator $\hat{c}_{GMM}$:

$$\hat{c}_{GMM} = (\hat{\Sigma}_{EX}^T \hat{\Sigma}_{EE}^{-1} \hat{\Sigma}_{EX})^{-1} \hat{\Sigma}_{EX}^T \hat{\Sigma}_{EE}^{-1} \hat{\Sigma}_{EY}.$$

We observe that the GMM estimator is identical to the two-stage least squares estimator of McDaid et al. [20] and Porcu et al. [10].

Therefore, both the least squares, regression of regression coefficients estimator and the two-stage least squares estimator are consistent estimators for the MVMR instrumental variable formula (12), and both belong to a more general family of GMM estimators.

## Causal effect estimation from simulated data

To test the accuracy of the least-squares (LS) and GMM estimators we performed simulations under a wide range of scenarios. In a first set of experiments, we considered exactly determined systems with normally distributed variables, where the LS and GMM estimators are identical (see Methods). As predicted by theory, the LS/GMM estimator is consistent with variance decreasing with increasing sample size in all simulations (Fig 2). In the literature, no simulations of MVMR with correlated instruments have been reported. We observed, perhaps surprisingly, very limited effect of increased instrument correlation on the variance of the causal effect estimates for pairwise correlations between 0.1 and 0.7, and even at correlation 0.9, the variance is only twice as large as at correlation 0.1 (Fig 2B and 2C).

Principal component analysis (PCA) has been suggested as a method to prune correlated SNPs in *cis*-MR studies by using a reduced set of uncorrelated linear combinations of SNPs as instruments instead [36]. We used the *multivariable principal component generalized method of moments* (MVMR PCA, function *mr_mvpcgmm* from the MendelianRandomization

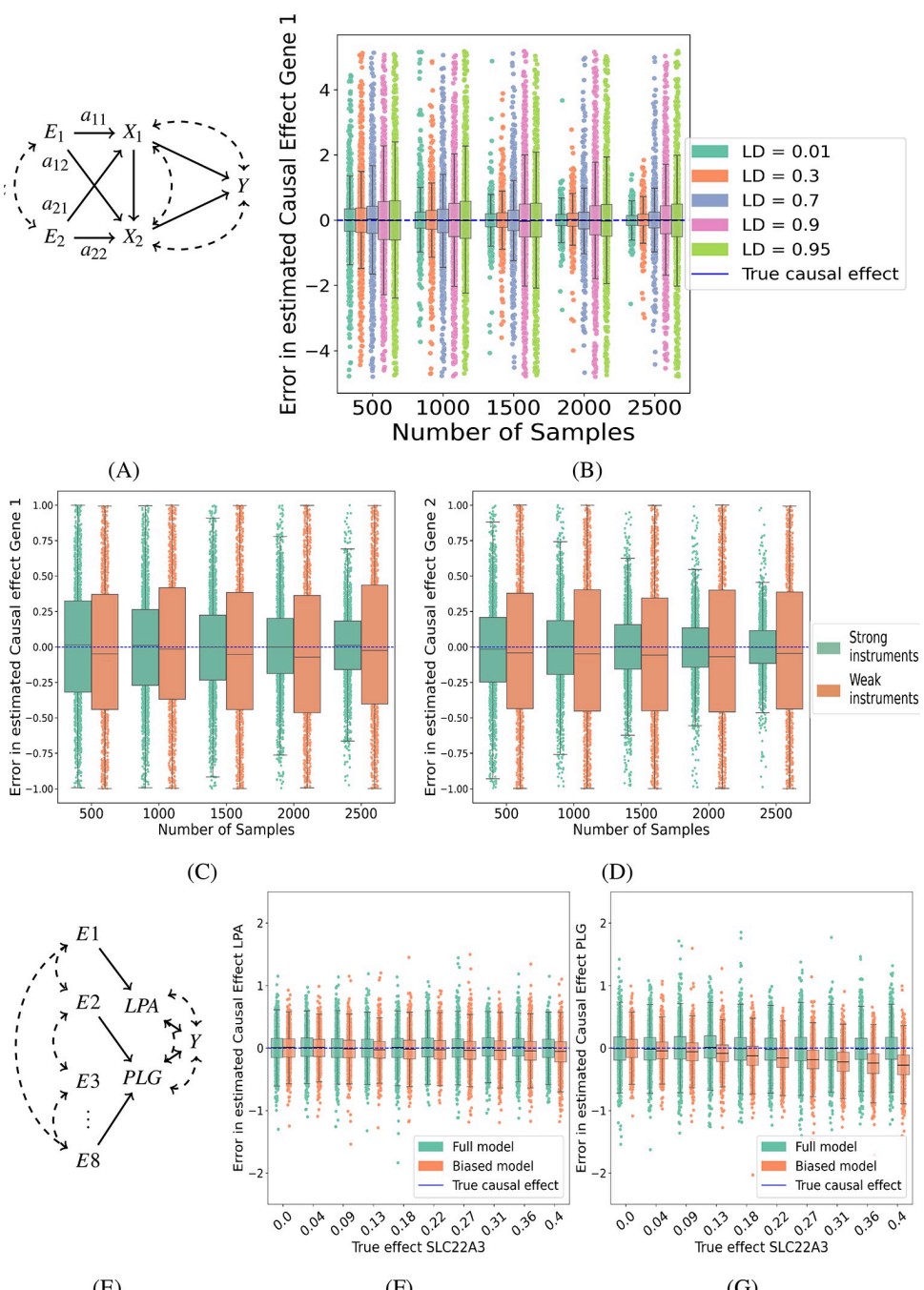

**Fig 2. Causal effect estimation from data simulating instrument correlations, instrument strength, and horizontal pleiotropy.** **(A)** Causal diagram for the simulation of two exposures $X_1$ and $X_2$ for an outcome $Y$, with two shared instruments $E_1$ and $E_2$ with pairwise covariance $\alpha$ and matrix of instrument effect sizes $A = [a_{ij}]$. **(B)** Difference $(\hat{c} - c)$ between estimated and true causal effect of $X_1$ (true effect size $c = 0.2$) on $Y$ across 1,000 independently simulated datasets for a range of sample sizes. **(C,D)** Distribution of the difference between estimated and true causal effects for $X_1$ (left; true effect size $c = 0.2$) and $X_2$ (right; true effect size $c = 0.6$) across 1,000 independently simulated datasets and a range of sample sizes comparing strong ($0.1 - 0.3$) vs weak ($0.001 - 0.01$) instruments. **(E)** False causal diagram used for inference where one causal exposure is missing. **(F-G)** Difference between estimated and true causal effect of $LPA$ (left, true effect size $c_{LPA} = -0.05$) and $PLG$ (right, true effect size $c_{PLG} = -0.27$) on $Y$ across 1,000 independently simulated datasets with sample size 2000 across a range of effect sizes of the hidden gene $SLC22A3$ mediating horizontal pleiotropic effects of the instruments on $Y$, showing both the estimates with the correct model (green) and the estimates with the hidden exposure model (orange). See Methods for simulation details.

package) [37] in our simple simulation and found that at an instrument of correlation value of 0.5 or more, the MVMR PCA method reported too few (that is, only one) effective degrees of freedom for causal effect estimation, despite the causal effects still being estimable.

We further tested the effect of weak instruments, that is, instruments with weak effects on the exposures (see Methods for details). As expected, weak instruments increase the variance in the causal effect estimates, more strongly at small sample sizes ($n = 500 - 1,000$) (Fig 2D and 2E). In general, the increase in variance due to weak instruments is not greater in the MVMR setting than in standard, univariate MR (S1B and S1D Fig).

In our next set of experiments, we used more realistic model parameters taken from real LD correlation matrices and eQTL and GWAS summary statistics from real loci in the human genome (see Methods for details). First we tested the effect of horizontal pleiotropy. Using real summary statistics from the *LPA-PLG-SLC22A3* locus, we simulated data from a model where all three genes (exposures) share causal instruments and are causal for the outcome *Y*, but we assumed only data for *LPA* and *PLG* was available (Fig 2F, see Methods for details). In other words, we assumed that the instruments have horizontal pleiotropic effects through the unmeasured gene *SLC22A3*. As expected, horizontal pleiotropy introduces bias in the estimated causal effects, although the bias remains modest (within the standard deviation of the estimates obtained from the complete model) upto a simulated effect size of around 0.15 of *SLC22A3* on the outcome (Fig 2G and 2H).

Next, we tested the effect of LD matrix bias, where the LD matrices in the eQTL and GWAS population differ from the reference LD matrix (e.g. from 1000 Genomes Project). For this purpose, we compared GMM estimates from data generated from the reference LD matrix and data generated from a biased LD matrix sampled from a Wishart distribution centred around the reference LD matrix (see Methods for details), using the reference LD matrix $\Sigma_{EE}$ in both GMM estimates (cf. Eq (10)). With a modest degree of LD matrix bias (Wishart distribution with 50 degrees of freedom), the bias in the causal estimates remained small (Fig 3B and 3C).

Next, we compared the LS and GMM causal effect estimators in over-determined models (more instruments than exposures). While both estimators are guaranteed to be consistent, the GMM estimator is the estimator with theoretically lowest variance (see Methods). However, across thousands of simulations with multiple sample sizes and for models of varying degree of over-determination, we did not observe any differences in variance between the LS and GMM estimators (Fig 3B and 3C).

We also simulated a two-sample setting where eQTL and GWAS LD matrices were sampled independently from two different Wishart distributions centred around the reference LD matrix, with different variance parameters to reflect the difference in sample size between eQTL and GWAS studies, and generated eQTL and GWAS summary statistics independently, with sample sizes representative of real eQTL and GWAS studies (see Methods for details). In the two-sample setting, even the effects estimated using the true LD matrix from the GWAS population are biased due to the LD matrix sampling variation between the eQTL and GWAS studies (Fig 3E). Unsurprisingly, the bias increased further when the reference LD matrix was used instead of the GWAS study LD matrix in the estimation (Fig 3F). In this setting, all estimators were biased until an eQTL and GWAS sample size combination of 4,000/140,000 was reached, which corresponds to the upper end of currently available sample sizes (Fig 3F); TWMR remained biased at all sample sizes, due to a hard-coded regularization factor (see Methods for details). We note that the causal effect bias in two-sample settings is not specific to MVMR or the use of correlated instruments, but can already be observed in univariate MR with a single instrument (S5 Fig).

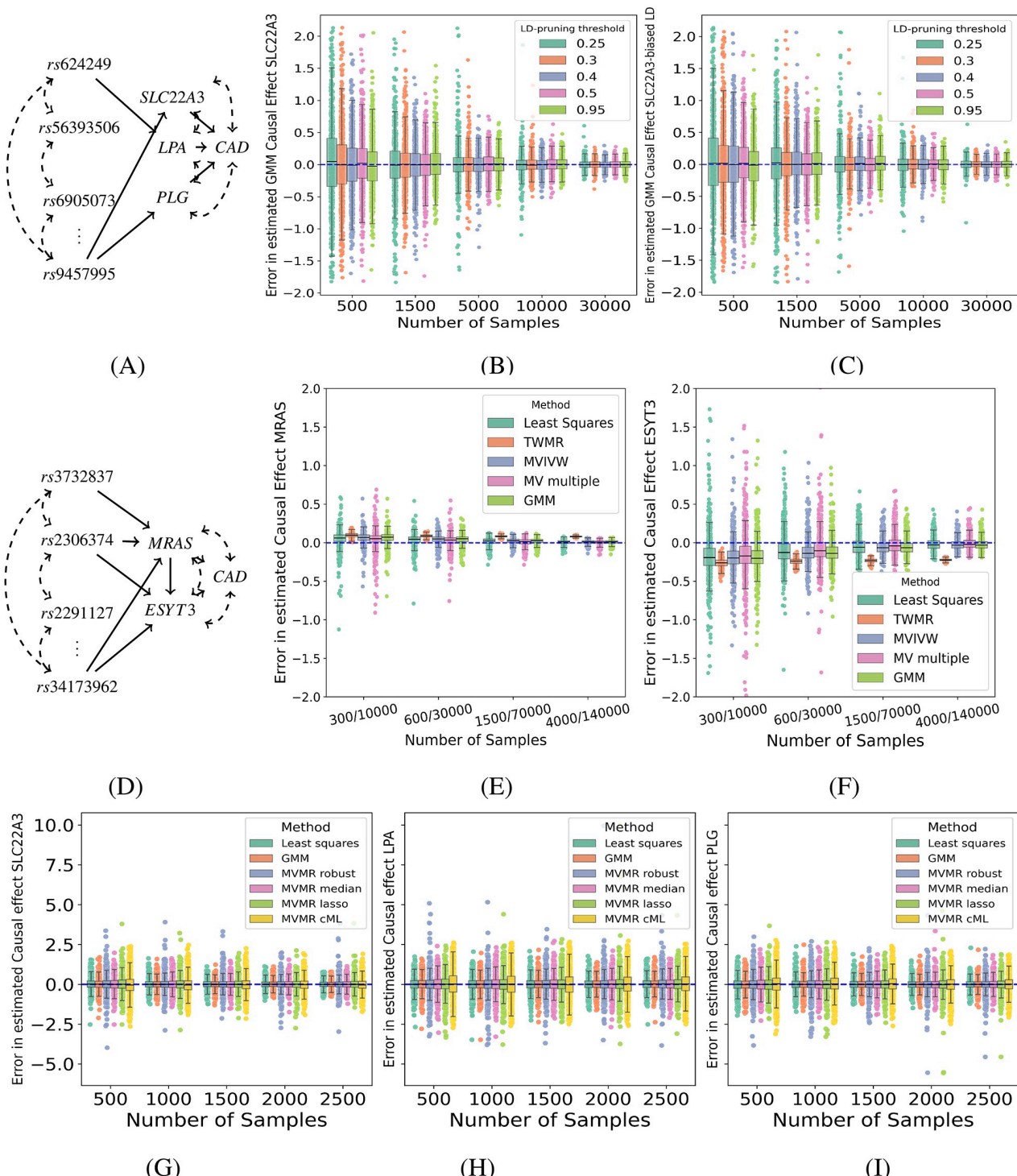

**Fig 3. Causal effect estimation from data simulating LD matrix sampling bias.** The first and second row show a causal diagram for the simulation of overdetermined systems where the number of causal variants was equal to the number of exposures (first row, three exposures; second row, two exposures) and other variants were only correlated with the exposures and outcome by LD. (**A**, **B**, **C**) Comparison between the GMM estimator using an unbiased (**B**) and biased (**C**) LD matrix for the causal effect of simulated *SLC22A3*, for different LD pruning thresholds (corresponding to models with 3, 4, 5, 6 and 7 instruments). (**D**, **E**, **F**) Comparison between the least-squares, GMM, TWMR, MV multiple (TwoSampleMR), and MVIVW (*Mendelian Randomization* package) estimators using independently sampled datasets of different sizes for exposures and outcomes to emulate a two-sample MVMR setting. (**G**, **H**, **I**) Difference between estimated and true causal effect of *SLC22A3* (left, true effect size $c_{SLC22A3} = 0.15$), *LPA* (center, true effect size $c_{LPA} = -0.05$) and *PLG* (right, true effect size $c_{PLG} = -0.27$) on $Y$ showing comparison between the least-squares, GMM, MVMR

Robust, MVMR Median, MVMR Lasso and MVMR cML estimators using simulated datasets from causal diagram (**A**). All distributions plots show the difference between estimated and true causal effects across 1,000 independently sampled datasets. See Methods for simulation details.

### Prediction of causal genes at GWAS loci for coronary artery disease

To test our method on real-world data, we predicted causal genes at 36 genome-wide significant loci for coronary artery disease (CAD) from the CARDIOGRAMC4D meta-analysis study [38] using eQTL data from seven vascular and metabolic tissues from the STARNET study [26] (see Methods for details). Results are presented here for the least squares estimator; the GMM estimates are predominantly consistent and are provided in the Supporting Data.

First, to get an understanding for the range of causal effect sizes one can expect for true causal genes, we identified 17 genome-wide significant CAD loci which had an effect on expression of a single candidate gene in a single tissue, that is, where no *cis*-regulatory pleiotropy is present and we can plausibly hypothesize that the true causal gene is known (Fig 4A). These genes included well characterized genes such as *PCSK9*, a risk gene for low-density lipoprotein cholesterol levels and CAD [39], whose visceral adipose (VAF)-specific association to CAD risk SNPs in STARNET was previously confirmed in independent data [26], and *GUCY1A3*, whose expression correlates with risk of atherosclerosis [40]. The absolute predicted causal effects on the standardized scale for these 17 genes ranged from 0.099 (*PCSK9* in VAF) to 0.34 (*GSTM3* in mammary artery (MAM)). We therefore considered an absolute causal effect size greater than or equal to 0.1 to be an appropriate threshold to define causal genes.

We confirmed the validity of this threshold using seven genome-wide significant CAD loci which had an effect on expression of a single candidate gene in at least two tissues in the STARNET data, that is, loci where we can plausibly hypothesize that the true causal gene, but not the tissue, is known. We used standard, univariate MR on a tissue-by-tissue basis and found that in all seven cases, the candidate gene had a predicted absolute causal effect size greater than 0.1 in at least one tissue (Fig 4B). These genes included *CDKN2B*, located in one of the most robust genetic markers for type 2 diabetes, CAD, and myocardial infarction [41]. GWAS SNPs in the *CDKN2B* locus were associated with expression of *CDKN2B* in aorta (AOR), subcutaneous fat (SF), and blood, with predicted causal effects $c = −0.75$, $−0.55$, and $0.5$, respectively.

We further computed p-values based on the standard errors of the estimated causal effects. As these p-values are based on one-sample estimates for the standard error and may not reflect variability in two-sample summary-data settings (see Methods for details), we conservatively set a Bonferroni threshold of $p < 0.05/150 = 3 \cdot 10^{-4}$, which exclude only five gene-tissue combinations with estimated causal effect $|\hat{c}| > 0.1$.

Having thus found a reasonable threshold to define causal genes for CAD, we analyzed 12 genome-wide significant CAD GWAS loci which had an effect on expression of multiple nearby genes in one or more tissues. Similar to Porcu et al. [10], we applied MVMR on a tissue-by-tissue basis using all genes with *cis*-eQTL associations in a locus in a given tissue as potential exposures in a causal diagram (cf. Fig 1B and 1E). We note that at only two of the 12 loci, it was possible to reduce the SNPs to a set of independent instruments using the LD threshold ($r^2 \leq 0.1$) used by Porcu et al. [10], while still retaining at least as many instruments as exposures. Hence an approach using correlated instrument sets is essential.

In total, 35 candidate causal genes with *cis*-eQTL associations were located in the 12 CAD GWAS loci with pleiotropic gene regulatory effects, of which 24 had a predicted absolute causal effect size greater than 0.1 in at least one tissue (Fig 4D). To analyze these causal genes

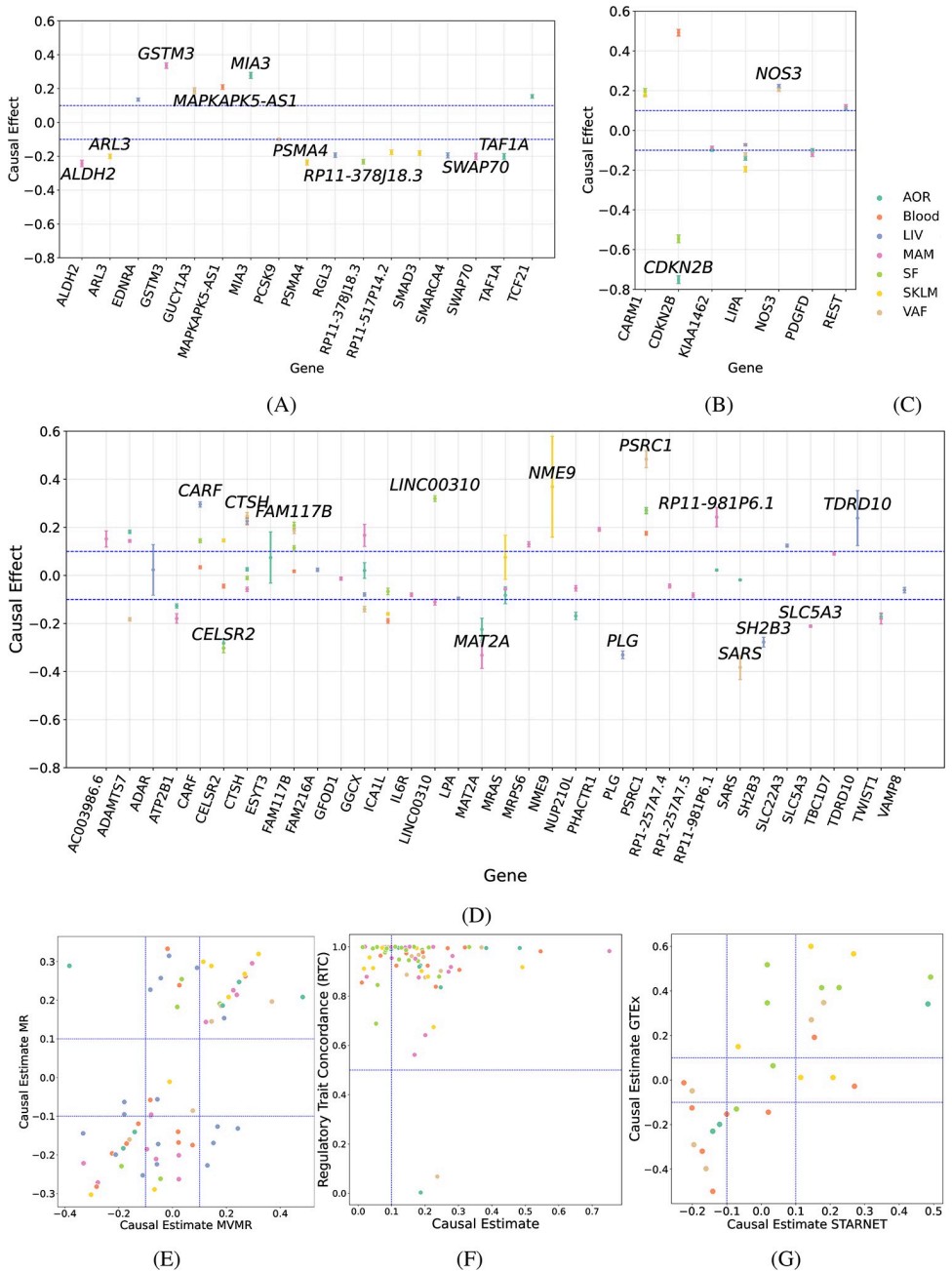

**Fig 4. Predicted causal genes and their tissue-specific effects on CAD at genome-wide significant CAD loci. (A)** Causal effects from univariate MR of genes at 17 CAD GWAS loci with effects on a single gene in a single tissue. **(B)** Causal effects from tissue-specific, univariate MR of genes at seven CAD GWAS loci with *cis*-regulatory effects on a single gene in multiple tissues. **(C)** Tissue color legend for all panels. **(D)** Causal effects from tissue-specific MVMR with correlated instrumental variable sets of genes at 12 CAD GWAS loci with *cis*-regulatory effects on multiple genes in multiple tissues. **(E)** Scatter plot of tissue-specific causal effect estimates from MVMR vs. univariate MR. **(F)** Scatter plot of tissue-specific causal effect estimates from MVMR vs. CAD regulary trait concordance (RTC) scores. **(G)** Scatter plot of tissue-specific causal effect estimates from MVMR in STARNET vs. GTEx.

more systematically, we compared the predicted causal effects from MVMR against the predicted causal effects from standard MR (Fig 4E) and regulatory trait concordance (RTC) scores [42] computed by Franzén et al. [26] (Fig 4F), two univariate methods that ignore the pleiotropic gene regulatory effects. We also conducted a replication analysis for loci with *cis*-associations in matching tissues in both STARNET and GTEx (Fig 4G, see Methods for details).

We observed a general trend where predicted causal genes using MVMR (predicted causal effect $|c| \geq 0.1$) were also predicted to be causal (using the same threshold) using univariate MR, had high RTC values, and had concordant effect sizes between STARNET and GTEx (Fig 4E–4H). However several genes with high univariate MR effects and high RTC values were *not* causal according to MVMR, suggesting that MVMR can indeed correct for likely false predictions from MR or colocalization analyses due to regulatory pleiotropy.

A clear example of a CAD GWAS locus where MVMR and MR give contrasting results is a locus centred around chr 6:12,901,441. In STARNET, the locus has *cis*-associations with the expression of *PHACTR1*, *GFOD1*, *TBC1D7*, *RP1–257A7.4*, and *RP1–257A7.5* in MAM (Fig 5A and 5B). In univariate MR analyses these genes have predicted causal effects ranging from 0.15 for *PHACTR1* to 0.31 for *GFOD1*, and a previous integrative genomics analysis lists all of them as candidate causal genes for this locus [43]. After pruning nearly identical SNPs (LD $r^2 \geq 0.95$), 12 instruments (with mutual LD ranging from $r^2 = 0.17$ to $r^2 = 0.86$) were available for conducting MVMR, resulting in a predicted causal effect of 0.19 for *PHACTR1*, with all other effects below the threshold of 0.1 in absolute value, suggesting that *PHACTR1* is the only causal gene in arterial tissue at this locus.

Another example is the locus centred around chr 15:79,141,784, where functional studies support a causal and proatherogenic role for *ADAMTS7* [44]. In STARNET, the locus has *cis*-associations with the expression of *ADAMTS7* in AOR, MAM, and VAF(Fig 5C and 5D). In all three tissues, the locus also has *cis*-associations with the expression of *CTSH*. After pruning nearly identical SNPs (LD $r^2 \geq 0.86$), 9, 25, and 10 instruments (with mutual LD ranging from $r^2 = 0.03$ to $r^2 = 0.94$) were available for conducting MVMR in AOR, MAM, and VAF respectively. In the arterial tissues, MVMR confirms that *ADAMTS7* is the most likely causal gene (predicted causal effects 0.18 and 0.14 vs. 0.025 and −0.058 for *CTSH* in AOR and MAM, respectively; Fig 5C). Replication in GTEx AOR tissue confirmed these results (predicted causal effects 0.18 and 0.009 for *ADAMTS7* and *CTSH*, respectively). By contrast, in VAF, both genes are predicted causal with effects in the *opposite* direction (predicted causal effects −0.18 and 0.25 for *ADAMTS7* and *CTSH*, respectively), the latter due to opposite eQTL effects of the same variants in VAF vs. AOR and MAM. No eQTL associations with either gene were available in GTEx in adipose tissue to confirm these results. The overall picture is further complicated by the fact that the locus also has *cis*-associations with the expression of *CTSH* in blood, SKLM, and SF, and with *PSM4* in SKLM. The absence of associations with *ADAMTS7* in these tissues implies that the association between the locus and CAD is automatically attributed to the other genes (Fig 5C). Given the overwhelming functional evidence for *ADAMTS7* [44], these results probably reaffirm the importance of having all true exposures included in the model (cf. Fig 2F–2H).

As a final example we consider the locus centred around chr 2:85,809,989. Previous analyses found a candidate causal SNP in this locus located in the promoter region of *MAT2A*, but since the SNP was associated with expression of multiple genes in this locus, a causal gene could not be predicted [45]. In STARNET, the locus has *cis*-associations with the expression of *MAT2A* in AOR and MAM. In both tissues, the locus also has *cis*-associations with the expression of *GGCX*, and both genes have previously been listed as candidate causal genes in arterial tissues [43]. After pruning nearly identical SNPs (LD $r^2 \geq 0.95$), three instruments (with mutual LD ranging from $r^2 = 0.69$ to $r^2 = 0.89$) were available for conducting MVMR. In both

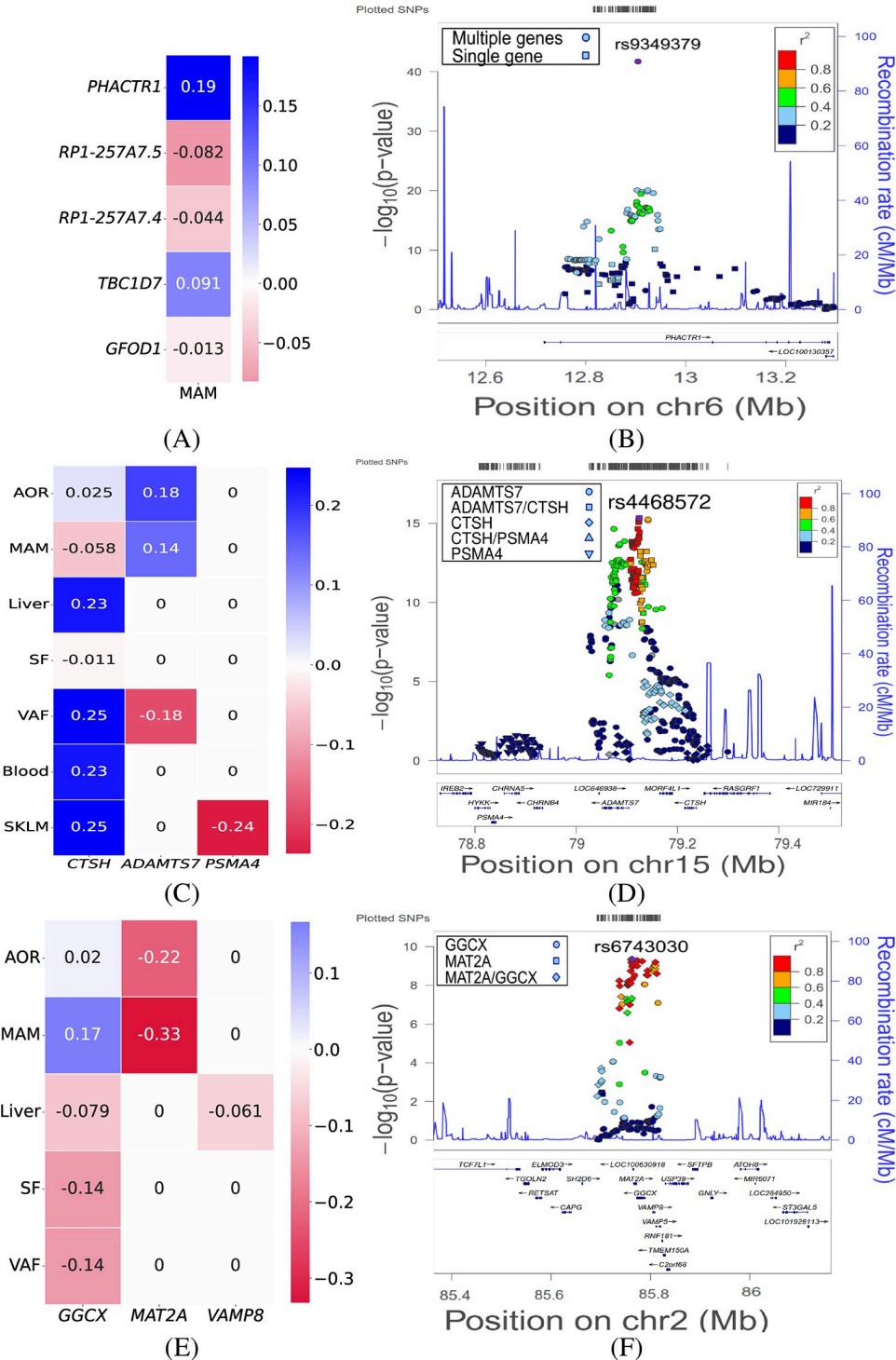

**Fig 5. Example CAD GWAS loci with pleiotropic gene regulatory effects and predicted causal genes.** Each row shows the predicted causal effects from tissue-specific MVMR of genes with *cis*-eQTL associations in a CAD GWAS locus of interest across seven vascular and metabolic tissues from the STARNET study (left) and the genetic architecture and *cis*-regulatory pleiotropy of the locus. The heatmaps show the causal effects on the liability on a standardized scale. Only tissues with non-zero *cis*-eQTL effects in the STARNET study are included. Values identically zero indicate gene-tissue combinations not included as exposures in the model (no *cis*-eQTL association in that tissue). The LocusZoom plots show for each SNP in the locus the GWAS $-\log_{10}(p-value)$ (y-axis) and associated *cis*-eQTL genes (symbol, see legends) in a selected tissue. (**A, B**) Locus centred on chr 6:12,901,44 with LocusZoom plot for

MAM. (**C**, **D**) Locus centred on chr 15:79,141,784 with LocusZoom plot for AOR. (**E**, **F**) Locus centred on chr 2:85,809,989 with LocusZoom plot for AOR.

AOR and MAM, MVMR suggests *MAT2A*, and not *GGCX* is the causal gene (predicted causal effects −0.23 and −0.33 vs. 0.02 and 0.16 for *GGCX* in AOR and MAM, respectively; Fig 5E and 5F). The overall picture is again less clear, since the locus also has *cis*-associations with the expression of *GGCX* and *VAMP8* in liver, and with *GGCX* alone in SF and VAF. Interestingly, in liver, neither gene is predicted causal (predicted causal effects −0.08 and −0.06; Fig 5E), while in the adipose tissues, the entire association between the locus and CAD is automatically attributed to *GGCX* (predicted causal effects −0.14 in both tissues; Fig 5E).

The preceding examples suggest that in fact two distinct forms of regulatory pleiotropy exist. A GWAS locus can be associated (in *cis*) to expression levels of *multiple* genes in the same locus in a tissue of interest, but can also be associated to *one or more* genes in *multiple* tissues. We attempted to conduct MVMR using all potential exposures (across genes and tissues) in a single model, in order to simultaneously predict the causal gene(s) and tissue(s) at loci of interest. However, conclusive results were often elusive, as both the determinant of the LD-matrix and the instrument strength matrix were very small, indicating that the number of causal variants in most loci is less than the often large number of exposures that must be accounted for in such a multi-tissue analysis.

An example of how multi-tissue MVMR can potentially differ from tissue-specific MVMR is given by the locus centred on chr 19:11,061,315. In STARNET, the locus has *cis*-associations with the expression of *CARMI* in SF and SKLM, and with *RGL3* and *SMARCA4* in liver. In tissue-specific analyses, *CARMI* was causal in SF ($c = 0.19$) and SKLM ($c = 0.18$) (univariate MR), while *RGL3* ($c = −0.19$) and *SMARCA4* ($c = −0.2$) were both predicted to be causal in liver (MVMR). Notably, the lead SNP rs35140030 was shared between *CARMI* in SF and SKLM, and another SNP in the locus, rs113718993 was shared between *CARMI* in SKLM and *SMARCA4* in liver. Since these SNPs were shared between tissues, and two additional instruments were available after pruning nearly identical SNPs (LD $r^2 \geq 0.95$), we conducted an additional MVMR analysis where exposures were *gene-tissue pairs*. This analysis predicted that (*CARM1*, SKLM) ($c = 0.18$) and (*RGL3*, liver) ($c = −0.19$) were the causal gene-tissue pairs, while (*CARM1*, SF) ($c = −0.02$) and (*SMARCA4*, liver) ($c = −0.01$) were not causal.

## Discussion

In this paper we studied if multivariate Mendelizan randomization (MVMR) can be used to identify causal genes at GWAS loci with pleiotropic gene regulatory effects, that is, GWAS loci associated with gene expression of more than one candidate gene in the genomic locus of interest. MVMR requires at least as many genetic instruments as the number of candidate genes included in the model, and the consensus in the field has been that these instruments must be independent. However, due to high levels of linkage disequilibrium between genetic variants located in the same genomic locus, it is usually impossible to identify such independent instrument sets.

Using the method of path coefficients we showed that the MVMR causal diagram with correlated instruments satisfies the *instrumental set condition*, a classical result by Brito and Pearl [24] that guarantees the identifiability of the direct causal effects of all exposures in the model on the outcome variable of interest. Moreover, the effects solve a regression of (univariate) regression coefficients, and the only requirement is that the set of instruments consists of (or tags) at least as many causal variants as the number of candidate genes included in the model.

We further showed that the standard two-stage least squares estimator of the causal effects in MVMR is part of a generalized method of moments (GMM) family of finite-sample estimators for the theoretical identification equation, as is a least squares estimator that simply replaces the theoretical univariate regression coefficients by their finite sample estimates, results which again do not depend on the presence or absence of instrument correlations.

Extensive simulations confirmed the validity and usefulness of these theoretical results. Most surprisingly perhaps was the finding that the variance in causal effect estimates remained small, even at high correlation values between the instruments. Moreover we saw no benefit in using the computationally more expensive two-stage least squares estimator instead of the simpler least squares estimator—both were unbiased and had identical variance in all our simulations. As expected, horizontal pleiotropy or misspecified LD matrices in the two-sample setting introduce bias in the estimated causal effects, as in all applications of MR, but using conservative simulation parameters, we observed convergence to the correct estimates at sample sizes at the upper end of what is currently available. Of note, in the one-sample setting, that is, when exposures and outcomes are simulated from the same LD structure, misspecified LD matrices (randomly sampled from a distribution centred around the true LD matrix) did not lead to noticeable bias or increased variance.

We applied our method to predict causal genes for CAD using eQTL data from seven vascular and metabolic tissues obtained from 600 coronary artery bypass grafting surgery patients in the STARNET study. While predicting CAD risk using expression data from CAD cases can be criticized, we have shown previously that more gene expression traits are associated with CAD risk loci in STARNET cases than in comparable disease-unspecific samples (e.g., GTEx) [26], that eQTLs inferred from CAD cases explain a large proportion of heritability of CAD risk [46, 47], and that variation in eQTL-associated genes and gene networks correlates with the extent of atherosclerosis in CAD cases [47, 48], suggesting that variation in gene expression and disease severity in CAD cases can indeed reflect underlying causal effects on CAD risk. This was further supported by the current finding that causal effects estimated from STARNET and GTEx samples for the same genes and tissues correlated well.

Our analysis of the STARNET data illustrated the importance of being able to apply MVMR with correlated instrument sets. Out of 36 genome-wide significant loci with *cis*-eQTL associations in STARNET, only 17 were associated with a single gene in a single tissue, 7 with a single gene in multiple tissues, and 12 with multiple genes in multiple tissues. A deeper look into the predictions at some of these loci with widespread regulatory effects showed both the strengths and limitations of applying MVMR in this context.

A relatively clear case was provided by the *PHACTR1* locus, where *PHACTR1* was predicted as the only causal gene in arterial tissue, even though GWAS variants in this locus had *cis*-eQTL associations to an additional four genes in the same tissue. The lead GWAS SNP in this locus is located in an intron of *PHACTR1*, and *PHACTR1* was one of two top candidate causal genes (together with *CDKN2B*) for CAD in an integrative genomics analysis that included STARNET data as well as orthogonal evidence [43]. Functional evidence also supports a causal role for *PHACTR1* [49], although recent results suggest effects of this locus on a distal gene *EDN1* may also play a role, and controversy about the true causal gene remains [50].

The *ADAMTS7* locus provided another important test case. Functional evidence strongly supports a causal role for *ADAMTS7* in CAD [44], and in the arterial tissues where the GWAS SNPs in this locus were associated with *ADAMTS7* expression, our method correctly predicted *ADAMTS7* as the causal gene, and not *CTSH*, another gene in this locus with *cis*-eQTL associations in the same tissues. However, in VAF, both genes were predicted causal, and when the method is applied to tissues where there are *no cis*-eQTL associations with *ADAMTS7*, other genes are predicted as causal genes. This latter result is consistent with what is seen in

simulations: if true exposures are missing from the model, the effect of the missing exposure is distributed over the available exposures in a manner that ensures consistency of the univariate instrument-exposure and instrument outcome covariances.

Throughout this paper, we have defined regulatory pleiotropy as the situation where the same genetic variants are associated to multiple genes in the same locus resulting in multiple paths by which the variants can affect downstream complex traits. Thus, accounting for (measured) regulatory pleiotropy using MVMR is different from methods that account for *unmeasured* horizontal pleiotropy. However, the fact that regulatory pleiotropy can extend over multiple cell types and tissues and that data from some potentially relevant cell types or tissues will almost always be missing, implies that the two concepts are interrelated: unmeasured regulatory pleiotropy likely is an important source of horizontal pleiotropy.

If data from multiple tissues are available, our results imply that MVMR must be run strictly speaking by including all gene-tissue pairs with *cis*-eQTL associations in a given GWAS locus as exposures in a single, large causal model in order to limit the risk of unmeasured pleiotropy. However, even though instrument sets can be highly correlated and the STARNET data contains only seven tissues, it was almost never possible to run such large models, as the requirement on the number of causal variants included in or tagged by the instrument set still stands. To overcome this limitation, we see two ways forward.

First, we should not consider MVMR to be a hypothesis-free method that can be run across all available tissue eQTL data to discover causal genes *de novo*. Instead prior knowledge of the relevant tissue and plausible candidate genes must be used to limit both the size of the model and the risk that true causal genes and gene-tissue combinations are missing from the model. For instance, integrative genomics analysis pipelines such as the ones developed by Brænne et al. [45] and Hao et al. [43] combine multiple eQTL and GWAS datasets with functional genome annotation and literature search to compile ranked lists of the most likely causal genes and tissues for an outcome of interest such as CAD. Using MVMR as an additional step in such pipelines to resolve causal effects at loci with eQTL effect on prioritized genes in prioritized tissues can provide additional causal evidence, as illustrated by the *PHACTR1* and *ADAMTS7* examples.

Another and more challenging solution would be to expand instrumental variable sets to the required size for models with a large number of exposures by including variants from outside the locus of interest (*trans*-acting variants) in the instrumental variable set. However such variants typically have small effects and are by definition indirect, such that care must be taken to exclude horizontal pleiotropic effects from upstream regulatory factors. Most likely, progress in this direction will require whole-network causal modelling in accordance with the omnigenic model [51].

We focused our analysis on genome-wide significant GWAS loci because the use of eQTL information to identify candidate genes is an important part of modern GWAS protocols [52] and because candidate genes at many GWAS loci have been analyzed using orthogonal data, providing some form of validation for our approach. Needless to say, our approach also could be applied to loci passing less stringent thresholds. More generally, the method could be integrated in existing methods to account for linkage disequilibrium and potential pleiotropic effects in transcriptome-wide association studies using statistical fine-mapping approaches [53–55]. These methods typically use sparsity-inducing constraints to predict outcome traits from a minimal set of most informative genes. Using results from a causal model such as considered here, either directly in an outcome prediction model, or indirectly as part of the sparsity constraints, could increase the probability that the minimal predictive gene set consists of truly causal genes.

In summary, through theory, simulation, and application on real-world data, we have shown that MVMR with correlated instrumental variable sets significantly expands the scope for predicting causal genes at GWAS loci with pleiotropic regulatory effects, but important challenges remain to account completely for the extensive degree of regulatory pleiotropy across multiple tissues.

## Supporting information

**S1 File. Supplementary Methods.** Contains 15 pages of supplementary methods. (PDF)

**S1 Fig. Causal effect estimation on simulated data of one dimensional systems. (A, C)** Causal diagrams for the simulation of one exposure $X$ for an outcome $Y$, influenced by one instrument $E$ with variable instrument strengths $a$ **(A)**, or influenced by $n \geq 2$ instruments $E_1$, ..., $E_n$ with instrument strengths $a_i$ **(C)**. **(B, D)** Distribution of estimated causal effects for $X$ (true effect size 0.3), showing distributions across 1,000 independently simulated datasets across a range of sample sizes under different simulation scenarios with varying instrument strengths. **(G)** Distribution of estimated causal effects for $X$ (true effect size 0.3) assuming the false diagram in **(F)** for inference when the true diagram that generated the data is in **(E)**. (PNG)

**S2 Fig. Causal effect estimation on simulated data for underdetermined and overdetermined systems. (A)** Causal diagram for the simulation of two exposures $X_1$ and $X_2$ for an outcome $Y$, influenced by three instruments $E_1, E_2, E_3$ where the number of causal variants is smaller than the number of cis-eGenes and other variants in the locus are merely associated by LD. **(B, C)** Distribution of estimated causal effects for $X_1$ **(B**, true effect size 0.2**)** and $X_2$ **(C**, true effect size 0.6**)** in the graph from **(A)**. **(C)** Causal diagram for the simulation of one exposure $X$ for an outcome $Y$, with $n \geq 2$ shared instruments $E_1, ..., E_n$ where the number of causal variants is greater than the number of cis-eGenes. **(E)** Distribution of estimated causal effects for $X$ (true effect size 0.3) from the graph in **(D)**. Panels **B**, **C**, and **E** show distributions across 1,000 independently simulated datasets across a range of sample sizes. (PNG)

**S3 Fig. Conditional F-statistic and causal effect estimation on simulated data.** The distribution of the conditional F-statistic and causal effect estimates are shown using two effect size ranges for weak and strong instruments: $0.001 - 0.01$ (weak) and $0.1 - 0.3$ (strong) in subplots **(A, B, C, D)**, and $0.001 - 0.03$ (weak) and $0.8 - 1.5$ (strong) in subplots **(E, F, G, H)**. The results are based on 2,000 simulations of over-determined systems for *ADAMTS7* (true effect size 0.15) where subplots **(A, E)** show the distribution of the conditional F-statistic using the *Mendelian Randomization* package and subplots **(B, F)** use the *MVMR* package. Subplots **(C, G)** display the distribution of causal effect estimates for the GMM estimator, while subplots **(D, H)** show the the distribution of the causal effect estimates for the mvivw estimator (using the *Mendelian Randomization* package). (PNG)

**S4 Fig. Standard error estimation and causal effect estimation comparison on simulated data of overdetermined systems. (A)** Distribution of estimated standard errors for *SLC22A3*, using the individual level data and exact form compared to using approximate form with summary level data. **(B)** Distribution of estimated causal effects for *SLC22A3* (true effect size 0.15) for the estimator GMM, showing distributions across 2,000 independently simulated datasets across a range of sample sizes using discrete instruments with randomly generated covariances

with real LD values from the *SLC22A3-LPA-PLG* locus.
(PNG)

**S5 Fig. Bias in univariate two-sample MR.** Difference $(\hat{c} - c)$ between estimated and true causal effect of X (true effect size c = 0.8) on Y across 20,000 independently simulated datasets for a range of sample sizes comparing bias of two sample MR vs one sample MR, in a simple ratio estimate with one instrument E. **(A)** Cov(*E*, *X*) and Cov(*E*, *Y*) estimated from the same sample, *Sample 1*. **(B)** Cov(*E*, *X*) estimated from sample, *Sample 1* and Cov(*E*, *Y*) estimated from sample, *Sample 2*. **(C)** Cov(*E*, *X*) estimated from sample, *Sample 2* and Cov(*E*, *Y*) estimated from sample, *Sample 1*. **(D)** Cov(*E*, *X*) and Cov(*E*, *Y*) estimated from the same sample, *Sample 2*. For simplicity, sizes of *Sample 1* and *Sample 2* were kept the same.
(PNG)

**S6 Fig. Standard error verification for STARNET.** Tissue-wise causal estimates from the Least squares estimator (red) and MVIVW estimator from the *Mendelian Randomization* R package, with standard errors (blue).
(PNG)

**S7 Fig. Estimated causal effects and their p-values.** Tissue-wise causal estimates from the MVIVW estimator from the *Mendelian Randomization* R package, with their corresponding $-\log_{10}(p-value)$. One outlier (*CDKN2B*) with estimated effect size -1.0 and p-value $<10^{-80}$ not shown.
(PNG)

**S8 Fig. Mean explained variance of the first principal component as a function of linkage disequilibrium (LD) *r* (correlation coefficient).** The red dashed line represents the theoretical expectation for the explained variance, calculated as $\frac{1+r}{2}$, which derives from the eigenvalues of the covariance matrix for two standardized variables with correlation *r*. The blue bars show the empirical mean explained variance of PC1 (Principal Component 1) obtained from simulations with a sample size of 2000 and 2000 repetitions.
(PNG)

**S9 Fig. Comparison of Type 1 error rate and power rate.** We estimate here the Type 1 error rate for the estimators Least Squares **(A)** and GMM **(C)** and Power rate for the estimators Least Squares **(B)** and GMM **(D)** from 2,000 independently simulated datasets for a fixed sample size of 2000, using discrete instruments with randomly generated covariances with real LD values from the locus on Chromosome 15:79124475 shared by genes *ADAMTS7* and *CTSH* in the MAM tissue.
(PNG)

## Author Contributions

**Conceptualization:** Mariyam Khan, Tom Michoel.

**Data curation:** Mariyam Khan, Sean Bankier.

**Formal analysis:** Mariyam Khan, Tom Michoel.

**Funding acquisition:** Johan L. M. Björkegren, Tom Michoel.

**Investigation:** Mariyam Khan, Adriaan-Alexander Ludl, Sean Bankier, Tom Michoel.

**Methodology:** Mariyam Khan, Adriaan-Alexander Ludl, Tom Michoel.

**Project administration:** Tom Michoel.

**Resources:** Mariyam Khan, Sean Bankier, Johan L. M. Björkegren.

**Software:** Mariyam Khan, Adriaan-Alexander Ludl.

**Supervision:** Tom Michoel.

**Validation:** Mariyam Khan, Tom Michoel.

**Visualization:** Mariyam Khan.

**Writing – original draft:** Mariyam Khan, Johan L. M. Björkegren, Tom Michoel.

**Writing – review & editing:** Adriaan-Alexander Ludl, Sean Bankier.

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
