## [Decision Letter · Decision Letter 0]

5 Mar 2024

Dear Dr Michoel,

Thank you very much for submitting your Methods entitled 'Prediction of causal genes at GWAS loci with pleiotropic gene regulatory effects using sets of correlated instrumental variables' to PLOS Genetics.

The manuscript was fully evaluated at the editorial level and by independent peer reviewers. The reviewers appreciated the attention to an important problem, but raised substantial concerns about the current manuscript. Specific concerns revolved around the model's inability to generate a p-value, the use of an effect size cutoff of 0.1 to declare significance, and a lack of demonstration regarding its effectiveness in controlling for horizontal pleiotropic effects. Based on the reviews, we will not be able to accept this version of the manuscript, but we would be willing to review a much-revised version addressing all these concerns. We cannot, of course, promise publication at that time.

If you decide to revise the manuscript for further consideration at PLOS Genetics, please aim to resubmit within the next 60 days, unless it will take extra time to address the concerns of the reviewers, in which case we would appreciate an expected resubmission date by email to plosgenetics@plos.org.

We are sorry that we cannot be more positive about your manuscript at this stage. Please do not hesitate to contact us if you have any concerns or questions.

Yours sincerely,

Xiang Zhou, Ph.D.

Academic Editor

PLOS Genetics

Michael Epstein

Section Editor

PLOS Genetics

Reviewer's Responses to Questions

**Comments to the Authors:**

Reviewer #1: Major comments:

The authors present a manuscript on cis MVMR, exploring the different options through simulations and application to real data summary statistics. The methods are well-established, nothing new is proposed, but the existing ones are compared through relatively simple simulation settings. Finally, they explore transcript to CAD causal effects in various tissues in an MVMR setting and compare it to univariable MR results. The results are plausible, the methods are sound, and the conclusions are reasonably insightful. While it is not a groundbreaking paper, it represents solid work. Below I provide some comments that could help to improve the manuscript.

There are some key points that would be great to address, which would bring in important novel aspects and increase the impact of the manuscript:

(i) the question of the accuracy of the LD – it is known that if we know the LD of the GWAS samples perfectly, we can include more SNPs and no need to prune aggressively, however, as the reference LD can deviate from the true LD in the GWAS (ref LD can be simulated from a Wishart distribution with the true LD), the discrepancy between the reference and target LD will cause instability in the estimation and substantial inaccuracy. This should be explored. This creates a bias-variance tradeoff: the more we prune, the lower the bias is, but the higher the variance is. However, if we prune little, the variance increases at the cost of increased bias due to LD mismatch.

(ii) No pleiotropy has been simulated for the instruments (could respect the INSiDE assumption). This would lead to overdispersion of the estimates, which could be explored.

(iii) A key difficulty in cis MVMR is the estimation of the estimator variance. The authors seem to completely ignore it.

(iv) Weak instrument bias in MVMR setting can be an important issue, I do not see a detailed discussion of the compromise between the conditional F-statistic and false discoveries. Fig Supp 1b shows no attenuation of causal effect estimates (and it is looking at univariable MR only?), which is surprising. This probably means that their simulation settings are not difficult enough (and would need to be tested with multiple exposures).

Methods perform very similarly (not surprising since they are somewhat equivalent), thus I’m not sure how important is to compare them against each other in the main text – I’d move cross-method comparisons [Fig 3] to the supplement.

There is a growing literature on cis-MR methods (see https://www.ncbi.nlm.nih.gov/pmc/articles/PMC7614127/ and methods within, especially https://pubmed.ncbi.nlm.nih.gov/28944551/), at least the latter one should be applied. The current simulations include a very small number of SNPs only (max 4), this should be scaled up to 15-20 to reflect more real life examples.

Lot could be cut/moved to the supplement in the sections “Causal effect identification in MVMR with correlated instruments” & “The generalized method of moments for causal effect estimation in MVMR with correlated instruments”

I’m not sure about the sample sizes used in the simulations. I see 500-2500, but typically the outcome GWAS would be much larger in terms of sample size. Were both sample sizes (eQTL & GWAS) varied?

In the real data application, some gold standard could be derived from orthogonal data to test whether MVMR pinpoints relevant genes? (E.g. drug target genes are more likely to be true hits or OMIM genes for monogenic forms of heart disease or CAD-related rarer diseases.)

Minor comments:

The font is too small in the legends (e.g. Fig2)

The locuszoom plots in Fig 5 are not really informative and hard to read. I’d remove them.

McDaid and Porcu et al do not propose two-stage least squares estimators, but two-sample MR estimators.

Reviewer #2: The authors introduce a MVMR method with correlated instrumental variables to identify the candidate causal genes at GWAS loci. Overall, the manuscript is well organized, especially with the rigorously theoretical justification. The simulations and real data analyses show good performance. However, I have some comments that the authors should be addressed, with details listed below.

Major Comments:

1. The authors should clarify the conceptual item “regulatory pleiotropy”, and elucidate how it differs from horizontal pleiotropy in traditional MR. Although utilizing molecular traits as exposures in MR can mitigate the risk of horizontal pleiotropy, concerns about pleiotropic bias remain. It would arise if variants were involved in different biological pathways relevant to the outcome, resulting in heterogeneous causal effect estimates. Intuitively, for the correlated IVs, only a small subset of them to be causal, while others will elevate the risk of horizontal pleiotropy. In addition, I suggested the authors to explore the impact of horizontal pleiotropy in the simulations.

2. Is it possible to obtain p-values using the derived least-square estimator or the GMM estimator, given that the p-value can also be obtained in MVMR and TWMR. It is quite uncommon to declare the significance only based on the absolute causal effect size greater than 0.1, rather than the p-value.

3. For the simulation with randomly generated instrument correlations, based on the Fig. 1A, what is the instrument strength a? Is it the effect size of the IV on the exposure? In addition, it seems to be confusing about the determinant of the instrument effect size matrix A. Why is the determinant of A directly associated with the instrument strength? For example, in a scenario with two IVs, if the effect size of the first IV on the exposures is 0.5 and 0.4, and for the second IV it is 0.4 and 0.5, the determinant of A is calculated as 0.5 * 0.5 - 0.4 * 0.4 = 0.09, indicating weak instruments. However, each instrument’s effect size is greater than 0.4, which can be view as the strong instrument.

4. For the simulation with instrument correlations derived from real LD matrices, the assumption that all eQTLs are causal for all genes may be unrealistic. The causal effect of eQTLs are more likely to be sparse. In addition, the two nearby genes are practically impossible to be the causal gene simultaneously. It is important to assess the performance of the proposed method when the genetic architecture is sparse.

5. In the real data application, the eQTL summary statistics is from CAD patients. It raises concerns about whether this data is suitable for evaluating the gene expression effect on CAD. The interpretation may be limited in the CAD patients rather than reflecting the general causal effect of a specific gene on CAD. In addition, why the authors focus on the genome-wide significant GWAS loci to perform MVMR analysis and use SNPs with GWAS p-value<5e-08 as IVs? In the traditional MR analysis, researchers often exclude SNPs significantly associated with the outcome to avoid violating the exclusion restriction assumption. Another consideration is that gene expression is estimated to mediate about 10–20% of trait heritability. It may exist some SNPs through alternative pathways affect the CAD in one specific loci.

6. Many TWAS fine-mapping method have proposed to model all pleiotropic genes in one genomic region, like FOCUS (Mancuso, N., Nat Genet, 2019), cTWAS (Zhao, s., Nat Genet, 2024), GIFT (Liu, L., Nat Genet, 2024). I wonder whether the theoretical framework proposed in this manuscript could be extended to these TWAS fine-mapping methods. The authors should, at least, discuss this.

7. Please clarify the impact of LD matrix calculation. Specifically, is the LD matrix derived from in-sample LD or an external reference panel? Is MVMR with correlated IVs sensitive to the difference between the LD reference panel and the GWAS data? What is the performance if we use the external reference panel, such as 1000 Genomic Project?

Minor Comments:

1. Please check the citation of figures, as there are multiple errors noted, particularly in Fig. 5B.

2. The description of the datasets is unclear. For example, the authors should explicitly state the sample sizes of exposure and outcome data used in both the simulation and real data applications.

Reviewer #3: The authors investigated an interesting topic of using correlated IVs in multivariable Mendelian randomization (MVMR) analysis. They showed the identification of causal effect in MVMR with the use of correlated IVs by applying the method of path coefficients. And they proposed a GMM estimator to estimate the causal effects, which has also been shown to be equivalent to two-stage least square estimator. I have concern in novelty and soundness of the current study.

1. My major concern is that as a causal inference method, the authors did not try to do any statistical inference on the causal estimator. The authors should propose a hypothesis testing procedure.

2. Related to the previous point, the current way to determine the presence of a causal relationship is based on some heuristic thresholds, e.g. 0.1 was used in the real data application. I strongly doubted the efficacy of such a heuristic method, which is based on the fact that the authors have already had a glimpse of the data and then determined the threshold. Again, a proper statistical testing should be used instead of using arbitrary thresholds.

3. In the real data application, I don’t understand why the authors selected IVs also based on the GWAS significance with the outcome CAD, while most MR studies would just select IVs simply based on their association with the exposure. Could the authors explain why? This may be problematic. First, it is very likely that pleiotropic variants would be included, if an IV is strongly associated with the outcome (perhaps even more significant than the IV-exposure association; for example, see Steiger’s filtering [1]); Second, selecting IVs based on the association with the outcome can introduce winner’s curse, and in the two-sample MR scenario, a upward bias in the causal estimator. This may have an impact in the current study design as the author used a threshold to determine causal gene.

4. As the authors also point out, their proposed GMM estimator is the same as TSLS, what is the major novelty/contribution in this study? The authors should further clarify this to strengthen their work.

5. In the simulation study, the authors should consider simulate IVs from a binomial distribution to reflect the nature of genotypes in MVMR. The authors mentioned in lines 300-302 that “…is not practical in our simulation, because the variables must be correlated according to a predefined LD matrix…” The authors may want to consider estimating the LD matrix based on their simulated genotypes, or even simulating another set of individuals to be used as a reference panel. Related to this, how did the authors estimate the LD matrix in their real data application?

6. It would be helpful if the authors can add some simple numeric examples based on a graph to illustrate their method, especially for Eqs. (1), (2).

7. In Figure S2 scenario (A) only one causal instrument is available for two exposures, which is described as an ‘underdetermined’ model, why is the performance of method still perfect? Is this contradict to page 10 line 402-404 “However, if the number of true causal variants is less than the number of exposures… does not help”.

Minor:

1. Typo: line 354, “LD greater than 0.0”

2. I am confused about “both p_i[V~Y] and p_j[E_j~V] point to V” in line 182. It would help if an example were provided.

[1] Hemani, Gibran, Kate Tilling, and George Davey Smith. "Orienting the causal relationship between imprecisely measured traits using GWAS summary data." PLoS genetics 13.11 (2017): e1007081.

**Have all data underlying the figures and results presented in the manuscript been provided?**

Reviewer #1: Yes

Reviewer #2: None

Reviewer #3: None

PLOS authors have the option to publish the peer review history of their article (what does this mean?). If published, this will include your full peer review and any attached files.

Reviewer #1: No

Reviewer #2: No

Reviewer #3: No

---

## [Decision Letter · Decision Letter 1]

6 Jul 2024

Dear Dr Michoel,

Thank you very much for submitting your Research Article entitled 'Prediction of causal genes at GWAS loci with pleiotropic gene regulatory effects using sets of correlated instrumental variables' to PLOS Genetics.

The manuscript was fully evaluated at the editorial level and by independent peer reviewers. While Reviewers 1 and 3 had only minor additional comments, Reviewer 2 felt that additional comparisons of your approach with existing MR approaches as well as justifications on the use of effect size threshold in place of p value threshold are warranted. Based on the reviews, we will not be able to accept this version of the manuscript, but we would be willing to review a revised version that suitably addresses Reviewer 2's concerns. We cannot, of course, promise publication at that time.

If you decide to revise the manuscript for further consideration at PLOS Genetics, please aim to resubmit within the next 60 days, unless it will take extra time to address the concerns of the reviewers, in which case we would appreciate an expected resubmission date by email to plosgenetics@plos.org.

We are sorry that we cannot be more positive about your manuscript at this stage. Please do not hesitate to contact us if you have any concerns or questions.

Yours sincerely,

Xiang Zhou, Ph.D.

Academic Editor

PLOS Genetics

Michael Epstein

Section Editor

PLOS Genetics

Reviewer's Responses to Questions

**Comments to the Authors:**

Reviewer #1: The authors made a great effort to address my comments and suggestions - I appreciate it.

I have only a few remaining minor comments left to be addressed, mostly regarding Figures:

1) Fig 2B: Y axis label should be replaced to "Bias of the estimated causal effect..."

2) Fig 2C: I'd remove this panel since it is common sense that the explained variance of the first PC for two variables with correlation r is (1+r)/2.

3) Fig 2D: The outer plot is not needed, since it only shows the outliers, the real information is in the inner plot. Also, weak instrument bias would bias the estimates towards the null, thus I'm surprised to see no bias in the effect estimation. Maybe the weak instrument bias is too small (i.e. the settings are too easy)?

4) The fact that the error in the LD does not lead to any bias is unexpected, but I guess it stems from the fact that the exposure and outcome effect estimates were simulated from the same LD structure. This could be added as a Discussion point.

Reviewer #2: I thank the authors for their efforts and hard work. I’m very happy with their replies to my previous comments and suggestions. However, I still have some comments that may need an additional attention.

1. To my knowledge, it is possible to use the rank inversion technique, or resampling methods to obtain the standard error. I wonder if these approaches would be better than the approximation from one-sample estimation.

2. It would be beneficial to compare the proposed least squares and GMM estimators to other existing MVMR methods, such as MVMR-Robust, MVMR-Median, and MVMR-Lasso (Grant AJ, Stat Med, 2021), MR-BMA (Zuber V, Nat Commun, 2020), and MVMR-cML (Lin Z, Am J Hum Genet, 2023).

3. Is there other literature also uses the threshold of causal effect size to declare the significance? Why use 0.1, and is there any literature to support this choice?

4. I think it is necessary to clarify the concept of regulatory pleiotropy and unmeasured pleiotropy. Besides the masking case, there exist instances where the IV affects the outcome without regulating gene expression.

5. I am confused about the statement: “We focused our analysis on genome-wide significant GWAS loci because the use of eQTL information to identify candidate genes is an important part of modern GWAS protocols [see e.g. Uffelmann et al., Nat Rev Methods Primers 1, 59 (2021)] and because candidate genes at many GWAS loci have been analyzed using orthogonal data, providing some form of validation for our approach. Needless to say, our approach also could be applied to loci passing less stringent thresholds or be integrated in TWAS fine-mapping methods”. Indeed, identifying the causal genes by integrating the eQTL dataset and GWAS dataset is important, and the two-sample MR require us to use the independent datasets of the exposures and the outcome. It would be better to perform the MVMR analysis without excluding the non-genome-wide significant GWAS loci and even excluding the genome-wide significant GWAS loci, and then compare the results to those in the previous manuscript. The authors' response about focusing on the genome-wide significant GWAS loci to perform the MVMR analysis is not convincing to me.

6. For the causal eQTLs, the authors sampled the instrument effect sizes randomly between 0.7 and 1.0. This setting is unrealistic.

7. I worry about the applicability of the proposed method. If the LD matrix is not invertible, which is common when using the LD matrix from the external reference data, how could we use the GMM estimator?

Reviewer #3: I would like to thank the authors for discussing all of my comments. I think the revision offers more clarity on the strengths and weaknesses of the proposed method. I have only a couple of further minor comments from the previous discussion.

1. On the reply to point 3 – I may misunderstand the following claim in the manuscript “We filtered these eQTLs by their p-values, retaining only those with p-values below 5 × 10−8 in the GWAS study.” Is the GWAS study here referred to the pQTL GWAS or the CAD GWAS? I previously thought this was referred to CAD GWAS, so I raised the questions in point 3. If the authors meant to refer to pQTL study, I suggest to rephrase this sentence.

2. I appreciate the authors’ effort to perform a more comprehensive simulation studies. It would be better to provide type-I error and power results besides point estimates.

3. Could the authors explain how to determine model identifiability in practice instead of theoretically? For example, is it possible to evaluate the three conditions in the data application? If not, some discussions would be helpful too.

**Have all data underlying the figures and results presented in the manuscript been provided?**

Reviewer #1: Yes

Reviewer #2: None

Reviewer #3: None

PLOS authors have the option to publish the peer review history of their article (what does this mean?). If published, this will include your full peer review and any attached files.

Reviewer #1: **Yes: **Zoltan Kutalik

Reviewer #2: No

Reviewer #3: No

---

## [Decision Letter · Decision Letter 2]

8 Oct 2024

Dear Dr Michoel,

Thank you very much for submitting your Research Article entitled 'Prediction of causal genes at GWAS loci with pleiotropic gene regulatory effects using sets of correlated instrumental variables' to PLOS Genetics.

The manuscript was fully evaluated at the editorial level and by independent peer reviewers. The reviewers appreciated the attention to an important topic but identified some concerns that we ask you address in a revised manuscript.

We therefore ask you to modify the manuscript according to the review recommendations. Your revisions should address the specific points made by each reviewer.

To resubmit, log into your Editorial Manager account and select the option 'Revise Submission' in the 'Submissions Needing Revision' folder.

Yours sincerely,

Xiang Zhou, Ph.D.

Academic Editor

PLOS Genetics

Michael Epstein

Section Editor

PLOS Genetics

Reviewer's Responses to Questions

**Comments to the Authors:**

Reviewer #1: Thank you for implementing/addressing my final comments/suggestions.

Reviewer #2: I have reviewed the revised manuscript, and I am pleased to see that the authors have addressed all of my previous comments and concerns satisfactorily.

Reviewer #3: I am happy with the authors' revision. I have one comment left: The supplementary file says that when implementing MVMR-cML, the K_vec is set as 5. However, if there is no invalid IV generated in the simulation, the K_vec should start from 0 in MVMR-cML. For example, it can be set as 0:5. I'd suggest the authors to double check the implementation and re-run the simulation.

**Have all data underlying the figures and results presented in the manuscript been provided?**

Reviewer #1: Yes

Reviewer #2: None

Reviewer #3: None

PLOS authors have the option to publish the peer review history of their article (what does this mean?). If published, this will include your full peer review and any attached files.

Reviewer #1: No

Reviewer #2: No

Reviewer #3: No

---

## [Editor Report · Decision Letter 3]

28 Oct 2024

Dear Dr Michoel,

We are pleased to inform you that your manuscript entitled "Prediction of causal genes at GWAS loci with pleiotropic gene regulatory effects using sets of correlated instrumental variables" has been editorially accepted for publication in PLOS Genetics. Congratulations!

Yours sincerely,

Xiang Zhou, Ph.D.

Academic Editor

PLOS Genetics

Michael Epstein

Section Editor

PLOS Genetics

Aimée Dudley

Editor-in-Chief

PLOS Genetics

Anne Goriely

Editor-in-Chief

PLOS Genetics

Comments from the reviewers (if applicable):

**Data Deposition**

http://datadryad.org/submit?journalID=pgenetics&manu=PGENETICS-D-24-00052R3

**Press Queries**

---

## [Editor Report · Acceptance letter]

4 Nov 2024

PGENETICS-D-24-00052R3 

Prediction of causal genes at GWAS loci with pleiotropic gene regulatory effects using sets of correlated instrumental variables 

Dear Dr Michoel, 

We are pleased to inform you that your manuscript entitled "Prediction of causal genes at GWAS loci with pleiotropic gene regulatory effects using sets of correlated instrumental variables" has been formally accepted for publication in PLOS Genetics! Your manuscript is now with our production department and you will be notified of the publication date in due course.

With kind regards,

Zsofia Freund

PLOS Genetics

On behalf of:
